# *Sex peptide receptor*-regulated polyandry modulates the balance of pre- and post-copulatory sexual selection in *Drosophila*

Juliano Morimoto [1,2,3], Grant C. McDonald[1], Emelia Smith[1], Damian T. Smith [4], Jennifer C. Perry[1,5], Tracey Chapman [4], Tommaso Pizzari[1] & Stuart Wigby[1]

Polyandry prolongs sexual selection on males by forcing ejaculates to compete for fertilisation. Recent theory predicts that increasing polyandry may weaken pre-copulatory sexual selection on males and increase the relative importance of post-copulatory sexual selection, but experimental tests of this prediction are lacking. Here, we manipulate the polyandry levels in groups of *Drosophila melanogaster* by deletion of the female *sex peptide receptor*. We show that groups in which the *sex-peptide-receptor* is absent in females (*SPR-*) have higher polyandry, and – as a result – weaker pre-copulatory sexual selection on male mating success, compared to controls. Post-copulatory selection on male paternity share is relatively more important in *SPR-* groups, where males gain additional paternity by mating repeatedly with the same females. These results provide experimental evidence that elevated polyandry weakens pre-copulatory sexual selection on males, shifts selection to post-copulatory events, and that the *sex peptide* pathway can play a key role in modulating this process in *Drosophila*.

[1] Department of Zoology, Edward Grey Institute, University of Oxford, South Parks Road, Oxford OX1 3PS, UK. [2] Department of Biological Sciences, Macquarie University, Macquarie, NSW 2109, Australia. [3] Programa de Pós-graduação em Ecologia e Conservação, Universidade Federal do Paraná, Curitiba 82590-300, Brazil. [4] School of Biological Sciences, University of East Anglia, Norwich Research Park, Norwich NR4 7TJ, UK. [5] Jesus College, University of Oxford, Turl Street, Oxford OX1 3DW, UK. Correspondence and requests for materials should be addressed to J.M. (email: juliano.morimoto@mq.edu.au)

Sexual selection is a powerful agent for driving evolutionary change, and arises from individual variation in fitness due to competition between members of one sex for access to fertilisation opportunities with the opposite sex[1–3]. Variation in male reproductive success was traditionally thought to be determined largely by the number of mates and by the number of offspring produced by those mates[4]. However, the discovery that the females of most species are polyandrous (mate with multiple males) has transformed our view of sexual selection[5–7]. It is now well established that polyandry can generate sexual selection on males after copulation (post-copulatory), by forcing the ejaculates of different males to compete for the fertilisation of the same ova[8–11]. Less clear is the way in which variable levels of polyandry in a population impact on the operation of pre- and post-copulatory episodes of sexual selection[12–14].

While several studies have attempted to elucidate the impact of polyandry on sexual selection in males, theoretical and empirical data have not reached a consensus. Theory suggests that increasing levels of polyandry in a population should, under certain conditions, weaken sexual selection on males by reducing the differences between males in the number of mates they obtain (their mating success). This may in turn reduce the strength of the correlation between mating success and the number of offspring sired (i.e. the Bateman gradient)[2,6,12,15,16]. Consistent with this expectation, empirical studies across a range of taxa have revealed that high polyandry is typically associated with low variance in male mating success—and thus potentially weak pre-copulatory sexual selection—indicating that opportunity for sexual selection is largely limited to post-copulatory processes[15,17–21]. Other studies, however, have suggested that pre-copulatory sexual selection is generally stronger than post-copulatory selection in polyandrous populations[22–25] and that increasing levels of polyandry accentuate—rather than weaken—sexual selection[26].

Two major factors are likely to underpin this lack of consensus. First, previous studies have not experimentally manipulated the level of polyandry in freely mating populations, which has prevented the unambiguous demonstration of the causal impact of polyandry on sexual selection. Second, estimates of the strength of sexual selection on particular components of male reproductive success (e.g. mating success) can be biased by covariance between different components (e.g. mating success and paternity share)[14,15,21,27]. These covariances are particularly relevant whenever the matings with females of varying levels of polyandry are non-randomly distributed across males[28,29]. For example, if males with high mating success tend to copulate with the least polyandrous females, an association inevitably forms between high mating success and low post-copulatory competition, which can strengthen net pre-copulatory sexual selection on males[28,30]. The potential for such non-random mating patterns to influence sexual selection can be empirically assessed by quantifying the structure of the matrix of matings between males and females (i.e. the mating matrix) within populations. However, many studies focus only on the average level of polyandry of a group, neglecting the way in which females of varying polyandry are distributed across males of varying polygyny[30]. Thus, an experimental approach is needed that directly manipulates polyandry levels, comprehensively measures the effect on pre- and post-copulatory sexual selection on males, and takes into account any variation in the mating matrix.

Here, we use such an experimental approach to manipulate the level of polyandry in an insect model system, the fruit fly *Drosophila melanogaster*. We studied 58 groups of flies—each consisting of four males and four females, which carried genetic eye markers and were painted for identification—in a set-up that allowed us to record the complete mating history and estimated

reproductive success of individual focal flies. In many insects, the degree of polyandry in a population is modulated by the responses of individual females to ejaculates received from their mates[31]. Typically, females undergo a refractory period of reduced sexual receptivity after mating, which suppresses polyandry and should decrease the intensity of sperm competition faced by males. In *D. melanogaster*, several female post-mating responses—including female receptivity to new matings, and fecundity—are strongly modulated by the *sex-peptide* (*SP*) pathway, whose function depends on the interaction between the male *SP* in the ejaculate, and the SP receptor (*SPR*) in the female's reproductive tract and nervous system[32–34]. In the absence of *SPR*, females more rapidly return to sexual receptivity after mating, resulting in substantially elevated polyandry[35]. We harnessed this effect to test how polyandry influences the operation of sexual selection on males. Given that there are multiple ways to measure sexual selection, each with their own potential limitations[36–39], we employed a comprehensive approach, utilising both univariate and multivariate selection gradients as well as variance decomposition. Specifically, we tested the hypothesis that elevated polyandry in groups where females lack *SPR* (i.e. *SPR*−) will weaken pre-copulatory selection on male mating success (the Bateman gradient[40]), and increase the relative contribution of post-copulatory selection (i.e., the proportion of the variance in male reproductive success explained by variance in paternity share), thereby altering the overall opportunity for sexual selection on males. Crucially, we also tested whether increasing levels of polyandry modulated the operation of sexual selection by influencing the relationship between the polygyny of males and the polyandry of their female mates, and thus the covariance between male pre- and post-copulatory performance (i.e. mating success and paternity share, respectively).

We first describe patterns of mating frequency in *SPR*− and control groups, showing that *SPR*− females are on average more polyandrous. We then compare the architecture of variance in male reproductive success between *SPR*− and control groups by assigning paternity of more than 11,000 female offspring. We show that, compared to control groups, *SPR*− groups are characterised by a reduced opportunity for pre-copulatory but not for post-copulatory sexual selection, due to reduced variance in male mating success. We follow this up by investigating how these differences between control and *SPR*− groups impact the strength of sexual selection on each component of male reproductive success, controlling for the confounding effects of other reproductive components, through a multivariate approach. We found that high mating rates in *SPR*− groups generated a negative relationship between the mating success (polygyny) of a male and the polyandry of his female mates, which promoted males with both high mating success and high paternity share. Together, these patterns indicate that polyandry weakens pre-copulatory sexual selection on male mating success in *SPR*− groups despite the buffering effect of a negative relationship between male polygyny and female polyandry in these groups. We further demonstrate that males can achieve high paternity share by repeatedly remating with their mates, and that males were under stronger post-copulatory sexual selection to do so in *SPR*− groups than in control groups. Finally, we show that these results are not explained by the reduced fecundity of *SPR*− females, suggesting instead that the observed differences in sexual selection on males between *SPR*− and control groups are caused by the experimentally elevated polyandry of *SPR*− females.

## Results

**Deletion of female *SPR* elevates polyandry.** We first assembled replicate groups of genetically marked *D. melanogaster* where

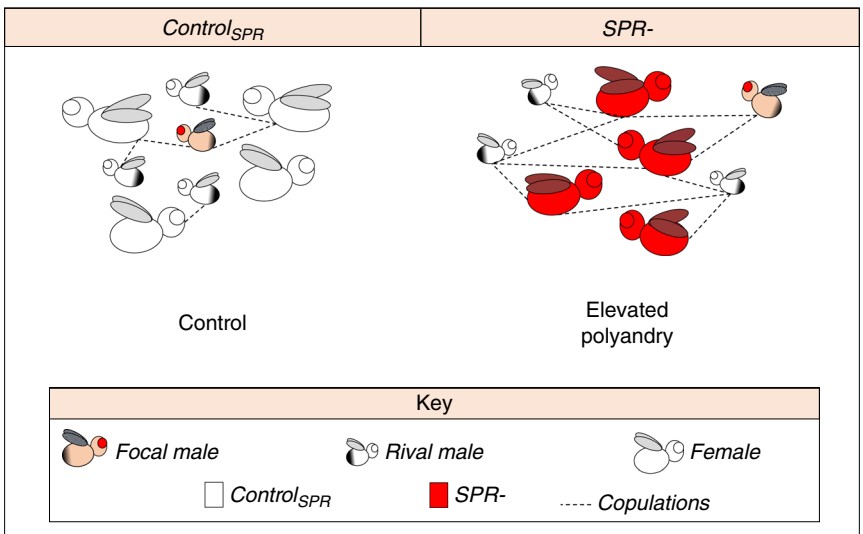

**Fig. 1** Schematic representation of the *SPR*− manipulation used to increase polyandry levels in freely mating populations (representative copulation patterns indicated by dashed lines). White cartoons refer to groups consisting of wild-type females (control$_{SPR}$); red cartoons refer groups consisting of *sex-peptide receptor*-lacking females (*SPR*−)

four males (three white-eyed rival males and one red-eyed focal male homozygous for the *spa* mutation) were kept with either four *SPR*− females or four control$_{SPR}$ females ($N = 29$ groups for each, 58 groups in total), all of which possessed the same genetic background (Fig. 1, Supplementary Figure 1). Flies were individually paint-marked, and we recorded all of their interactions during observation periods of 4 h over each of 4 successive days, followed by egg-laying periods for each individually housed female. This experimental design allowed us to identify all individual matings, determine the reproductive success of individual females and assess the total number and proportion of daughters sired by the focal and rival males with each of their female mates (the genetic markers we used allowed us to assign the paternity of daughters but not sons: see Methods), generating an accurate estimate of reproductive success for focal males. As expected, we found that polyandry levels were significantly elevated in *SPR*− groups relative to control$_{SPR}$ groups (Supplementary Figure 2A). On average, the single focal male in each group accounted for approximately one-third of the matings observed, while his three rivals accounted for the remaining two-thirds (F test: Focal vs. Rivals: $F_{1,112} = 20.050$, $p < 0.001$), with no statistically significant differences between the patterns observed in *SPR*− and control$_{SPR}$ treatments (F test: Focal vs. Rivals × Treatment: $F_{1,110} = 1.220$, $p$ value = 0.272; Supplementary Table 1). These mating frequencies indicate that, although the individual focal males on average tended to perform better than individual rival males, this effect did not differ between treatments, and thus there is no evidence of a systematic bias (Supplementary Table 1). Crucially, the data also show that the rivals collectively provided effective competition for focals, and generated ample scope for variation in mating success—and thus potentially sexual selection—between focal males, across groups. As expected based on polyandry levels, focal males in the *SPR*− groups had on average both higher overall mating frequencies and more female mates than control$_{SPR}$ group focal males (Fig. 2a, b). We also found that focal males in the *SPR*− groups sired a significantly lower proportion of the offspring produced by each of their mates compared to focal males in the control$_{SPR}$ group (Fig. 2c). Despite these striking differences in mating rate, mating success and paternity share between treatments, and slightly reduced female productivity in *SPR* groups (Supplementary Figure 2B), absolute reproductive success did not differ significantly between focal males in the *SPR*− and control$_{SPR}$ treatments (Fig. 2d). Thus, the paths to reproductive success for focal males likely differed between the *SPR*− and control treatments, but the reproductive outcomes, in terms of productivity, were equivalent.

**Deletion of *SPR* alters variance in male mating success.** We tested the prediction that higher polyandry levels should reduce the opportunity for pre-copulatory sexual selection on male mating success (the standardised variance in mating success, $I_S$), while maintaining the opportunity for post-copulatory sexual selection on paternity share (the standardised variance in paternity share, $I_P$[2,15,18]; Table 1). Our results were fully consistent with these patterns: $I_S$ was significantly lower in the *SPR*− treatment than in the control$_{SPR}$ treatment while there was no difference in $I_P$ (Fig. 2e, f, Supplementary Table 2). Based on these results, we expected the opportunity for total sexual selection (i.e. standardised variance in total male reproductive success, $I$; Table 1) to be relatively low in *SPR*−, due to the low variance in mating success ($I_S$), which was not compensated by variance in paternity share ($I_P$; see e.g. ref. [15]). We found that $I$ was reduced considerably in the *SPR*− treatment compared to the control$_{SPR}$, which is in the direction consistent with our prediction, but this effect was not statistically significant (Fig. 2g, Supplementary Table 2).

To reveal what factors mediate the effects of the *SPR* on sexual selection on males, we dissected male reproductive success into its constituent parts: mating success (number of mates, $M$), the average productivity (rate of offspring production) of a male's mates ($N$) and share of paternity ($P$) of the offspring produced by a male's mates. We expected the relative role of $P$ to increase when polyandry levels were high (i.e. in the *SPR*− groups), because of the expected increase in importance of post-copulatory processes in predicting male reproductive success, and conversely, we expected the role of $M$ to decrease, because pre-copulatory processes should become less important. Our results are fully consistent with this prediction: $M$ accounted for nearly half of the variance in $T$ of focal males in the control$_{SPR}$ groups, whereas $M$ accounted for <14% of the variance in $T$ in the high polyandry *SPR*− groups. Conversely, $P$ was the main source of variance in focal male $T$ in the *SPR*− groups, accounting for more than 63% of the variance, whereas it accounted for only a little over 43% of

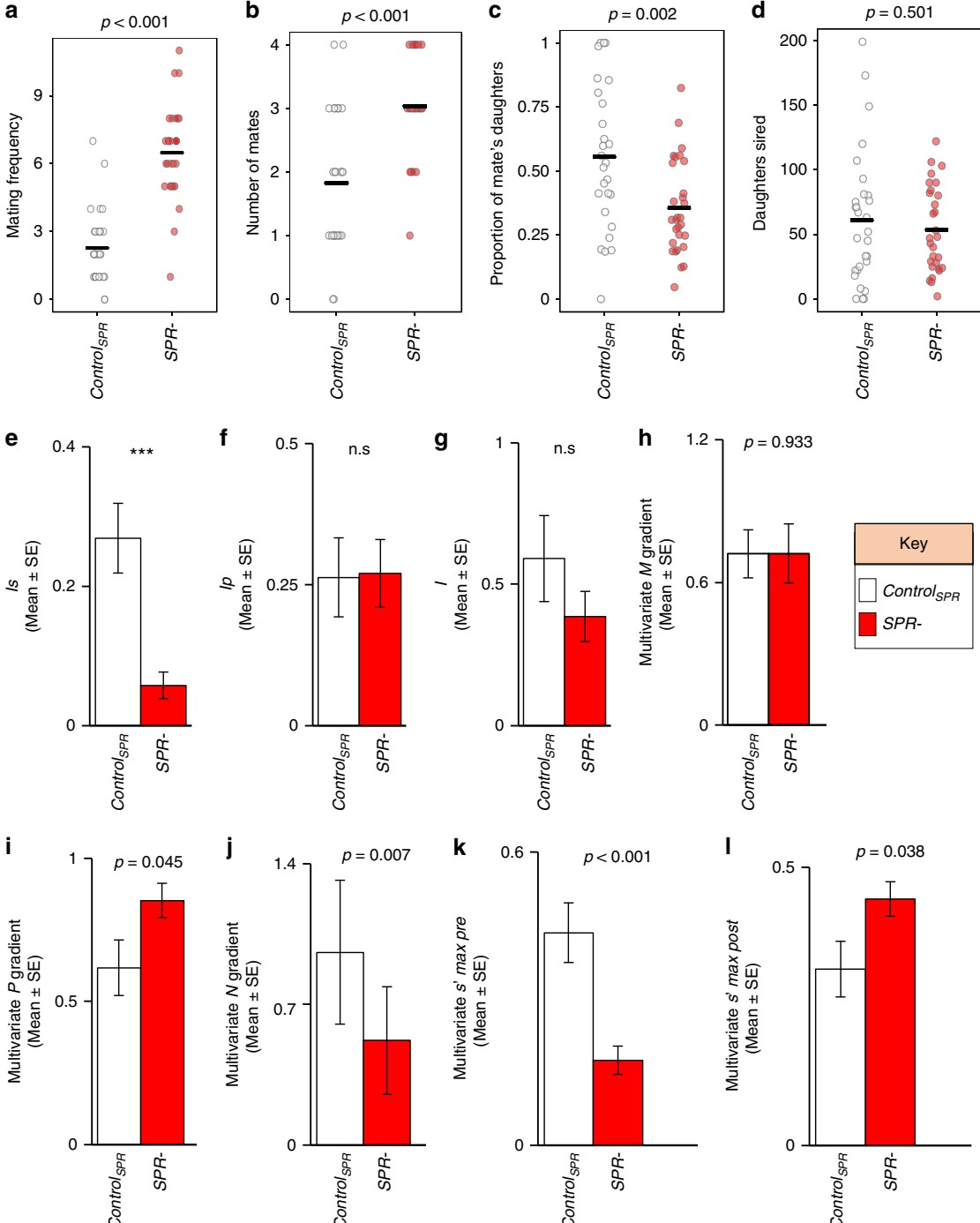

**Fig. 2** SPR effects on polyandry and the operation of sexual selection on males. **a** Focal male total mating frequency. **b** Focal male mating success (number of unique mates); **c** proportion of mate's daughters sired by focal males; **d** total number of daughters sired by the focal males; **e** the standardised variance in focal male mating success ($I_S$) (the opportunity for pre-copulatory sexual selection), ***: non-overlapping 95% bootstrap confidence interval; **f** the standardised variance in focal male siring success (opportunity for post-copulatory sexual selection, $I_P$), n.s.: overlapping bootstrap confidence intervals; **g** the standardised variance in offspring (daughters) sired by the focal males (opportunity for selection, $I$), n.s.: overlapping bootstrap confidence intervals; **h** the multivariate gradient of focal male mating success ($M$) on reproductive success ($T$); **i** the multivariate gradient of focal male paternity share ($P$) on reproductive success ($T$); **j** the multivariate gradient of the productivity ($N$) of the mates of focal males and reproductive success ($T$); **k** the maximum standardised multivariate pre-copulatory sexual selection differential index (multivariate s'max (pre)); **l** the maximum standardised multivariate post-copulatory sexual selection differential index (multivariate s'max (post)). Sample sizes: $N = 27$ for control$_{SPR}$ and $N = 29$ for $SPR-$ treatment. **a–d** Horizontal black line represents the mean of the data; p values were obtained from $F$ tests from GLM models. **e–l** Error bars in refer to the standard error of the mean (SE). Note the differences in scale of the y axis in each panel. White colour refers to control$_{SPR}$ treatment; red colour refers to the $SPR-$ treatment. **h–l** P values obtained from Student's $t$ tests from multivariate linear regressions

**Table 1 Sexual selection indexes and formulas used for their calculations**

| Sexual selection index | Abbreviation | Formula |
|---|---|---|
| Opportunity for selection | $I$ | $var(T)/mean(T)^2$ |
| Opportunity for pre-copulatory sexual selection | $I_S$ | $var(M)/mean(M)^2$ |
| Opportunity for post-copulatory sexual selection | $I_P$ | $var(P)/mean(P)^2$ |
| Univariate pre-copulatory ($M$) gradient (i.e. Bateman gradient)[a] | $\beta_{SS}^{Uni}$ | $T \sim \beta_{SS}^{Uni} \times M + covariates$ |
| Univariate post-copulatory ($P$) gradient[a] | $\beta_P^{Uni}$ | $T \sim \beta_P^{Uni} \times P + covariates$ |
| Multivariate pre-copulatory gradient[a] | $\beta_{SS}^{Multi}$ | |
| Multivariate post-copulatory gradient[a] | $\beta_P^{Multi}$ | $T \sim \beta_{SS}^{Multi} \times M + \beta_P^{Multi} \times P + \beta_N^{Multi} \times N + covariates$ |
| Multivariate mate productivity gradient[a] | $\beta_N^{Multi}$ | |
| Mean $P$ on repetitive matings with the same female[a] | Repetitive matings with the same females | $P \sim Matings$ with the same females $+ covariates$ |
| Multivariate maximum pre-copulatory index[b] | Multivariate s'max (pre) | $\beta_{SS}^{Multi(var)}$ |
| Multivariate maximum post-copulatory index[b] | Multivariate s'max (post) | $\beta_P^{Multi(var)}$ |
| Univariate pre-copulatory Jones' index[b] | Univariate Jones' index (pre) | $\beta_{SS}^{Uni}\sqrt{I_S}$ or $\beta_{Ss}^{var}$ |
| Univariate post-copulatory Jones' index[b] | Univariate Jones' index (post) | $\beta_P^{Uni}\sqrt{I_P}$ or $\beta_P^{var}$ |
| Sperm competition intensity | SCI | $\dfrac{1}{\frac{1}{M_i}\left(\sum_j^M \frac{1}{k_j}\right)}$ |
| Sperm competition intensity correlation | SCIC | $SCI \sim SCIC \times M$ |

$T$ focal male reproductive success, $M$ focal male mating success, $P$ focal male paternity share, $N$ focal male's mate productivity. Covariates include vial fecundity (except for the repetitive mating gradient) and replicate. $\beta_x^{(var)}$ = variance-standardised gradient of $x$, where $x$ is either $M$ (pre-copulatory) or $P$ (post-copulatory). $\beta_x^{Uni}$ or $\beta_x^{Multi}$ univariate and multivariate mean-standardised gradients of $x$, where $x$ is either $M$, $P$ or $N$. For the SCI calculation, $M$ is the mating success of the focal $i$th male and $k_j$ is mating success of the $j$th female that mated with the focal male
[a] Mean standardisation as $x/\bar{x}$
[b] Variance standardisation as $x - \bar{x}/sd(x)$, where $x$ is either $M$, $P$ or $N$

**Table 2 Decomposition of the variance in male reproductive success**

| Var–cov components | Control$_{SPR}$ | % | SPR− | % |
|---|---|---|---|---|
| var($T$) | 2.188 | | 1.106 | |
| var($M$) | 1.129 | 44.7 | 0.151 | 13.6 |
| var($P$) | 1.106 | 43.7 | 0.704 | 63.6 |
| var($N$) | 0.517 | 20.8 | 0.164 | 14.8 |
| cov($M$, $P$) | −0.378 | 14.9 | 0.027 | 2.4 |
| cov($M$, $N$) | 0.239 | 11.6 | 0.064 | 5.8 |
| cov($N$, $P$) | 0.223 | 8.8 | 0.235 | 21.2 |
| $D$ | −0.648 | | −0.239 | |

Relative contributions of mating success ($M$), paternity share ($P$) and mate productivity ($N$) to the variance in male offspring siring ($T$). Delta method of variance decomposition[15,27,76]

the variance in $T$ for control$_{SPR}$ groups (Table 2). The contribution of $N$ to the variance in $T$ was more limited and more uniform across treatments, although slightly greater in our control$_{SPR}$ groups (ca. 21% and 15% in control$_{SPR}$ and SPR−, respectively; Table 2). The low contribution of $N$ in both control$_{SPR}$ and SPR− treatments is expected given that flies were reared under standardised conditions (see Methods) designed to minimise variation in productivity between individual females, while the slightly lower contribution of $N$ in SPR− groups likely reflects the effect of increased polyandry, which reduces variance in $N$ across focal males.

Together, these results are broadly consistent with the idea that elevated polyandry, caused by the lack of SPR in females, erodes the opportunity for pre-copulatory, but not post-copulatory, sexual selection on males. Some previous studies, mostly in socially monogamous birds, have suggested that polyandry can increase the operation of sexual selection in males, by increasing the variance in mating and reproductive success in males, due to extra-pair copulations skewing the mating success in favour of a

few males[22–24,26]. However, our results indicate that an experimental increase in polyandry, in an already moderately polyandrous system, such as that of *D. melanogaster*, can erode the variation in male mating success. Collectively, these empirical results provide a direct experimental confirmation of recent theoretical arguments that the introduction of polyandry into a monogamous mating system can increase sexual selection, whereas further increases in polyandry in an already polyandrous mating system may reduce sexual selection[15,19,29].

**Deletion of the SPR weakens pre-copulatory sexual selection.** Next, we used a multivariate approach to estimate the strength of sexual selection on the components of male reproductive success $M$, $P$ and $N$. We investigated the gradient of the relation between $M$, $P$ and $N$ and male reproductive success ($T$), while controlling for the other components and their covariances[27]. Based on the findings about the opportunity for sexual selection above, we expected that the removal of the SPR in females would lead to a reduction in the gradient between $M$ and $T$ in males, and would increase the gradient between $P$ and $T$, relative to controls. This is because sperm competition caused by polyandry should diminish the reproductive returns of gaining additional mating success ($M$), and should place more emphasis on paternity share ($P$)[2,6,12,16,28]. To test these predictions, we measured the relative strength of pre- and post-copulatory sexual selection through the mean-standardised multivariate $M$, $P$ and $N$ gradients. This approach revealed no difference in the $M$ gradients of focal males between SPR− and control$_{SPR}$ treatments (Student's $t$ test: $t_{46} = -0.083$, $p = 0.933$, Fig. 2h). We also investigated the strength of sexual selection using the widely used mean-standardised regression between mating and reproductive success (univariate Bateman gradient, which does not control for $P$, $N$ or covariances). This approach again revealed no significant difference in the $M$ gradient of focal males between SPR− and control treatments (univariate Bateman gradient: $t_{52} = 0.689$, $p = 0.488$; Supplementary Table 3). These results are consistent with the results of

the multivariate $M$ gradient and suggest that males gain similar increments in reproductive success per unit increase of mating success across $SPR-$ and control$_{SPR}$ groups. However, we found a significantly steeper gradient of $T$ on $P$ for focal males from the $SPR-$ treatment compared with focals from the control$_{SPR}$ (Student's $t$ test: $t_{46} = 2.054$, $p = 0.045$) (Fig. 2i). Conversely, the gradient of $T$ on $N$ was significantly lower in the $SPR-$ compared to control$_{SPR}$ treatments (Fig. 2j). Taken together, these results suggest that, when gradients were mean standardised, the experimental manipulation of $SPR$ had an impact on $P$ and $N$ selection gradients, but not on $M$ selection gradients, as focal males in both treatments gained a similar reproductive benefit per unit increase of relative mating success (i.e. they had similarly strong $M$ gradients). A possible explanation for these results is that standardising gradients by the population mean does not take into account systematic changes in variance. Because elevated polyandry was associated with reduced variance in male mating success, mean-standardised Bateman gradients may be less sensitive to the effects of polyandry-induced changes in variance[29,41]. We tested this idea by investigating the effect of treatment on variance- (rather than mean-) standardised $M$ and $P$ gradients. We thus measured the univariate and multivariate s'max, which combines information on the variance in male mating success ($I_S$) and the relationship between $T$ and $M$ measured above, to estimate the maximum potential strength of pre-copulatory selection on phenotypic traits[41,42] (Table 1, Supplementary Table 4, see also Supplementary Figure 3). This approach revealed that the univariate pre-copulatory s'max (also known as the Jones' index) was halved in the $SPR-$ treatment compared with control$_{SPR}$ (Supplementary Figure 3) and that the multivariate pre-copulatory s'max index was significantly weaker in the $SPR-$ treatment relative to control$_{SPR}$ treatment (Fig. 2k, Supplementary Table 4). We also measured the comparable univariate and multivariate s'max for post-copulatory $P$ (see Table 1), and found that the univariate post-copulatory s'max (equivalent to a post-copulatory Jones' index) was almost doubled in the $SPR-$ treatment compared with control$_{SPR}$ (Supplementary Figure 3). The multivariate post-copulatory s'max index was significantly stronger in the $SPR-$ treatment relative to control$_{SPR}$ treatment (Supplementary Table 4, Fig. 2l), which is primarily a result of the steeper gradient of $T$ on $P$, since the other component of the index—the standardised variance in paternity share $I_P$—did not differ between treatments (see Fig. 2f). Together, these results provide experimental evidence that, consistent with previous correlational studies, high polyandry levels can reduce the maximum strength of pre-copulatory sexual selection while increasing the maximum strength of post-copulatory sexual selection[2,6,12,15,16].

**SPR affects the relationship between polyandry and polygyny.** A possible factor contributing to the effects of $SPR$ on sexual selection could be alterations to the patterns of mate sharing. This could occur if, for example, males who obtain many mates mated non-randomly with the most polyandrous females and therefore faced the highest intensity of sperm competition[28,30,43–46]. To address this possibility, we first estimated each male's sperm competition intensity (SCI)—which reflects the average polyandry of a male's mate—and examined the relationship of SCI with male reproductive success ($T$). We found a negative correlation between male reproductive success ($T$) and SCI, while controlling for male mating success, in both control and $SPR-$ treatments ($\Delta^T$control$_{SPR}$: Estimate: $-1.226 \pm 0.373$, $p = 0.003$; $\Delta^T SPR-$: Estimate: $-1.361 \pm 0.425$, $p = 0.004$). We also found a negative correlation between male paternity share ($P$) and male SCI, again in both control and

$SPR-$ treatments (control$_{SPR}$: Estimate: $-3.153 \pm 0.208$, $p < 0.001$; $SPR-$: Estimate: $-2.620 \pm 0.196$, $p < 0.001$). Together, these results confirm that male reproductive success and fertilisation success are negatively associated with SCI. Next, we measured the relationship between the mating success of a male (i.e. his polygyny) and the polyandry level of his female mates (SCI), to calculate the SCI correlation (SCIC)[28] (see Methods). Negative SCIC values indicate that the most polygynous males tended to mate with the least polyandrous females (resulting in a negative covariance between $M$ and the intensity of sperm competition). Positive values suggest the most polygynous males tend to mate with the most polyandrous females and thus face more sperm competitors. We found that SCIC were drastically different between $SPR-$ and control groups: $SPR-$ groups were characterised by negative SCIC values, while control groups were clustered around 0 (Treatment: SCIC $\pm$ SE; control$_{SPR}$: $0.028 \pm 0.078$; $SPR-$: $-0.124 \pm 0.055$). This indicates that, in the $SPR-$ treatment, less polygynous males were restricted to sharing more polyandrous female mates, whereas males with higher mating success had on average a more exclusive relationship with their female mates. In control groups, this relationship was absent. This reveals that some degree of pre-copulatory sexual selection on male mating success is retained in $SPR-$ groups, because highly polygynous males in this treatment may enjoy relatively less sperm competition compared to their rivals, than is the case for the highly polygynous males in control groups. In other words, polyandry would have reduced pre-copulatory sexual selection on male mating success more in the absence of the negative relationship between male polygyny and female polyandry in the $SPR-$ treatment. Randomisation tests showed that our observed SCIC values were not more extreme than could be expected by chance (i.e. random mating), given the distribution of mating pairs in the population, for either $SPR-$ or control treatments (Fig. 3a; $SPR-$ treatment: $p_{rand} = 0.194$, control: $p_{rand} = 0.97$; Fig. 3a). Therefore, the most parsimonious explanation for the negative SCIC in our $SPR-$ treatment is via increased saturation of the matrix of potential male and female mates (the mating matrix). As more pairs copulate in the more polyandrous $SPR-$ treatment, the mating matrix of pairwise matings between males and females becomes increasingly saturated, which tends to restrict the range of possible SCIC values toward negative values (Fig. 3a, ref. [28]). Thus, our results show that the negative relationship between male polygyny and female polyandry, emerging from the saturation of the mating matrix in the $SPR-$ treatment, means that males with high mating success obtain more reproductive success by facing reduced sperm competition.

**Males in $SPR-$ groups mate repeatedly with the same female.** In many species, a male can potentially defend his paternity in the face of sperm competition by remating repeatedly with the same female (repetitive mating), to increase the representation of his sperm within the female reproductive tract[15,16,21,25]. We found that focal males in the $SPR-$ treatment mated significantly more often with the same female than did focal males in control$_{SPR}$ groups (Fig. 3b, Supplementary Table 5). We tested whether this behaviour was a likely contributor to the stronger post-copulatory sexual selection for males in the $SPR-$ treatment described above, by fitting a general linear model of paternity share ($P$) on the average number of repetitive matings. We found a significantly steeper gradient between standardised $P$ and repetitive matings in the $SPR-$ groups compared to control$_{SPR}$ groups (Fig. 3c, Supplementary Table 5), which suggests that remating with the same female at high frequency was more strongly favoured by post-copulatory sexual selection under high polyandry. For completeness, we also investigated the gradient of adjusted paternity

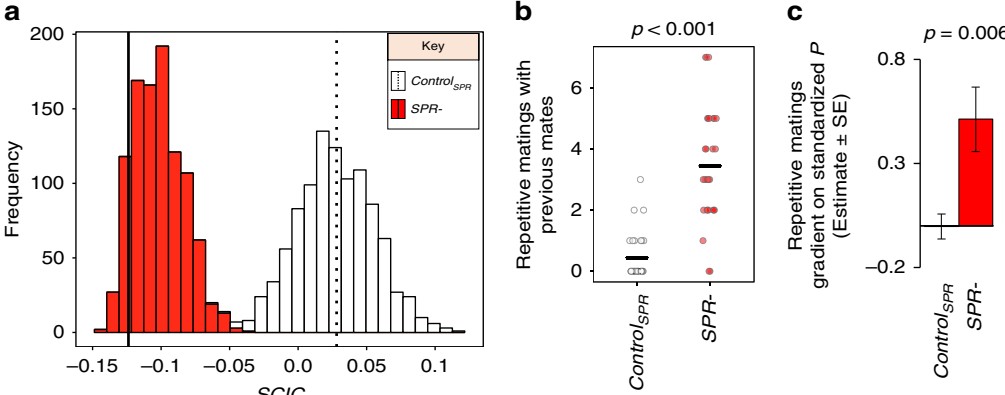

**Fig. 3** Mechanisms by which polyandry modulates pre- and post-copulatory episodes. **a** Stacked frequency plot showing the simulated null distributions of sperm competition intensity correlation (SCIC) values generated from 1000 randomisations of empirical mating data for *SPR−* and control populations. Vertical lines highlight the observed SCIC value in our experiment. Dashed for *SPR−* treatment; solid for controls. Treatments differ in the range of SCIC values generated by randomisations, as a result of differing levels of polyandry between treatments. Observed values do not lie outside the range of values expected under the null hypothesis of random mating. **b** Average focal male's repetitive mating with same female (mean ± SE). *P* value obtained from a *F* test of a GLM model. Horizontal black line represents the mean of the data. **c** Repetitive mating gradient on mean-standardised *P*. *P* value obtained from a Student's *t* -test of a Linear Regression Model. Sample sizes: $N = 27$ for control$_{SPR}$ and $N = 29$ for *SPR−* treatment. Estimate—the estimate of the gradient. Error bars in **c** refers to the standard error of the mean (SE). White colour refers to control$_{SPR}$ treatment; red refers to the *SPR−* treatment

share on repetitive matings, where we adjusted the paternity share of focal males to the realised level of sperm competition as in Devigili et al.[47] (see Supplementary Table 6). Similar to our previous findings, the results showed a positive gradient of repetitive matings on adjusted paternity share in the *SPR−* treatment, an effect that was not observed for control$_{SPR}$ (Supplementary Table 6). Thus, the gradient of remating on adjusted paternity share was steeper in the *SPR−* treatment than in the control$_{SPR}$, although this difference did not reach statistical significance (Supplementary Table 6). Nonetheless, these results broadly support the idea that post-copulatory sexual selection on repetitive matings with the same female is stronger in the *SPR−* treatment than in the control$_{SPR}$.

**Reduced female productivity and changes in sperm storage.** In addition to modulating female receptivity to remating—and hence the degree of polyandry—*SPR* affects other post-mating changes in females, including feeding, fecundity and the associated use of stored sperm[34,48–50]. Changes in fecundity and sperm release in females could affect the dynamics of sperm competition, potentially influencing the variance in male paternity share and estimates of sexual selection[15,16]. *SPR−* females displayed a small but significant reduction in productivity in our study (Supplementary Figure 2B). This is likely due to the fact that *SPR−* females produce fewer eggs than controls after mating[34], although the strength of this effect seems to be much weaker in studies in which female multiple mating is permitted than in single-mating assays[51,52]. Reduced female fecundity decreases male fertilisation opportunities, so could potentially influence sexual selection on males, for example, by increasing variance in male paternity share, $I_P$. The fact that $I_P$ did not differ across *SPR−* and control$_{SPR}$ treatments (i.e. Fig. 2f) suggests that the reduction in female productivity in our *SPR* experiments did not influence our estimates of sexual selection on males. However, to verify this independently, we performed an additional experiment based on the same experimental design, but using females with ablated insulin-like peptide-producing median-neurosecretory cells (mNSC-ablated females) and appropriate controls[53,54] in place of *SPR* females (see Supplementary Tables 3–8 and Supplementary Figure 4). mNSC-ablated females

have dramatically reduced fecundity and productivity but, in our experiments, did not differ from controls in polyandry levels (Supplementary Figure 5). We found that, in the presence of mNSC-ablated females, there was no significant change in $I_P$ nor any effect on other sexual selection indices compared to control$_{ablated}$ (Supplementary Figures 5–8, Supplementary Tables 3–7). The results therefore support the idea that a small reduction in productivity cannot explain the broad suite of changes to sexual selection on males seen in the *SPR−* treatment.

We then investigated whether changes in sperm storage previously observed in *SPR−* females[50,51] could have influenced our estimates of the strength of sexual selection. To investigate this, we reanalysed data from a previous study that examined the temporal pattern of sperm storage and paternity share of competing males mating with both control$_{SPR}$ and *SPR−* females[51]. In Smith et al.[51] females mated exactly twice (i.e. polyandry was constant), and the reproductive success of the first and second males was assessed (Supplementary Figure 9). We compared the standardised variance in paternity share $I_P$ (Table 1 and Methods) across males mating with *SPR−* and control$_{SPR}$ females, applying a focal male approach to match the methods in our own experiments. We randomly sampled one focal male per twice-mated female in the data set and calculated $I_P$ for males mated to *SPR−* and control$_{SPR}$ females. $I_P$ was calculated based on the reproductive success of females after mating with both mates. We repeated this process 1000 times to generate a distribution of $I_P$ values for both treatments. Differences between the estimates of $I_P$ would suggest that the levels of post-copulatory competition in *SPR−* and control$_{SPR}$ females are not comparable, and that changes in sexual selection on males are potentially a consequence of the *SPR*'s effect on post-copulatory competition patterns other than its effects on polyandry. However, the comparisons show that estimates of $I_P$ males in the *SPR−* female and control treatments were not significantly different (Treatment: mean (95% CI); control$_{SPR}$: 0.838 (0.586, 1.141); *SPR−*: 0.784 (0.533, 1.082), Supplementary Figure 9). These results provide further support for the conclusion that it is the increase in polyandry in *SPR−* females, rather than any other effects of the *SPR*, that drives the changes in the strength and opportunity of sexual selection on males observed in our study. It

is important to note that although the *SPR−* construct also lacks four other genes of unknown function[34], which could have some off-target physiological and behavioural effects, previous studies have shown that the effects of *SPR* knockout used here are similar to those observed using more specific manipulations of *SPR*[50,55,56]. Thus, even though we cannot completely rule out any off-target effects of our *SPR−* construct, available evidence suggests that the deletion of *SPR* per se is likely the major factor generating the polyandrous phenotypes observed in our experiments.

## Discussion

Our findings provide experimental clarification for the long-standing debate over the implications of polyandry for the operation of sexual selection on males. The results demonstrate that experimental manipulations to elevate polyandry in an already moderately polyandrous species weaken, rather than strengthen, sexual selection on the ability of males to gain mates. The data also clearly showed a concomitant increase in the importance of post-copulatory strategies, such as repeated remating with the same female, in determining male reproductive success. Collectively, these results indicate that increasing polyandry acts to saturate the matrix of possible male–female mating pairs (i.e. mating matrix), reducing pre-copulatory sexual selection on male mating success while promoting post-copulatory sexual selection on male traits that increase and defend paternity share, such as for example favouring a male remating with the same female. Importantly, these results are unlikely to have arisen solely by the saturation of the mating matrix. In a recent study using computer simulations, McDonald and Pizzari[29] demonstrated that the reduction of the importance of pre-copulatory sexual selection was primarily driven by elevated group polyandry, and that the saturation of the mating matrix only accelerated this process. This effect was true over a range of group sizes (10 males:10 females, 100 males:100 females) larger than our experimental groups (4 males:4 females), and was observed with saturation of the mating matrix as low as 2%, highlighting the primary role of increased polyandry—not the saturation of the mating matrix—in reducing the importance of pre-copulatory sexual selection[29].

Our results have broad relevance, and in particular shed light into the operation of sexual selection in species with high-remating rates and group-structured populations. The small groups used our experiments clearly do not capture the full range of population structures found in *D. melanogaster* in nature. However, by manipulating the *SPR*, we have experimentally induced a shift in group mating dynamics away from the natural behaviour of our *D. melanogaster* base population, and instead exposed fundamental properties of the causal relationship between polyandry and sexual selection processes, which are also relevant to other populations or species. Sexual competition in small groups and remating within the same pairs is common and an important component of sexual competition in many taxa. Male remating with the same female has been documented in numerous taxa including insects, gastropods, birds and mammals[15,17,21,57–67], and has been shown to function as a sperm competition defensive strategy[16,21,26]. Within the genus *Drosophila*, *D. hydei* females remate within an hour and can remate >3 times in a single day[68], which may create the potential for same-mate remating, and as a result might lead to some of the sexual selection processes seen in our *SPR−* treatment. While our understanding of wild *D. melanogaster* mating patterns remains limited, there is experimental evidence that females can display preferences for previous mates[69] and that males defend recently mated females from competitors to ensure paternity[70]. Together,

these results suggest that some degree of same-female remating may be sexually selected also in some *D. melanogaster* populations. Ultimately, the fitness returns to males associated with remating with the same female vs. mating with new females will be modulated by factors such as group size, sex ratio and the window of opportunity for intersexual interactions. Future empirical studies should seek to clarify the way in which these factors modulate polyandry-mediated changes to sexual selection on males. More generally, however, our results are likely to be broadly relevant for sexual selection processes resulting from mating patterns typically found across a wide range of taxa.

The fact that the changes in polyandry in our experimental *D. melanogaster* groups were brought about through modifications in the broadly conserved *SP* pathway[34] suggests a potentially common molecular pathway underpinning the relationship between polyandry and sexual selection in insects. Natural genetic variation in the SPR pathway gives rise to variation in remating rates in *D. melanogaster*[71], raising the possibility that this pathway could drive sexual selection on males in wild fruit fly populations. In some insect species, females are monandrous and thus post-copulatory sexual selection on males is absent, while in others, females can mate several times a day[72], promoting intense post-copulatory sexual selection and weak pre-copulatory selection[73]. Uncovering whether these inter-specific differences are related to the *SP* pathway, or other genetic pathways that mediate the degree of polyandry, will facilitate our understanding of the mechanisms underlying their evolution via sexual selection. More broadly, given that polyandry is ubiquitous[5], understanding the drivers of variation in polyandry both within and between populations and species, and the role of this variation in sexual selection, remains an important challenge for future laboratory and field studies.

## Methods

**Fly husbandry**. Experiments were conducted at 25 °C, in a non-humidified room on a 12:12 light–dark cycle. Flies were reared at standard density and fed on standard Lewis medium fly food. Fly food vials for adults were supplemented with ad libitum live yeast granules. The genetic lines used in this study were (i) a recessive *w^1118^* allele, which is a loss-of-function allele for the *white* gene in the X chromosome that confers white eyes in homozygotes; (ii) a deficiency, *Df(1) Exel6234*, on the X chromosome which deletes the *SPR* gene and four other genes of unknown function[34] and contains an insert of a *white+* transgene that partially rescues the *white* phenotype to produce orange (heterozygote) or red (homozygote) eyes[52]; (iii) a *sparkling^poliert^* allele (*spa*), located on the fourth chromosome, which produces a rough-looking eye phenotype[74], (iv) a *UAS-reaper* (*UAS-rpr*) transgene and (v) an *InsulinP3-GAL4* (*InsP3GAL*) driver, which when crossed with *UAS-rpr* (to give *UAS-reaper > InsP3GAL*) ablates insulin-like-peptide-producing mNSCs (henceforth 'mNSC-ablated' females;[53]). A *white^Dahomey^* stock was created by serially backcrossing *w^1118^* into the Dahomey genetic background. *SPR*, *UAS-rpr* and *InsP3GAL* were backcrossed into the *white^Dahomey^* genetic background for at least five generations, and *spa* was backcrossed into Dahomey, again for five generations: thus, all the flies used in these experiments, including those used in the previous study whose data we reanalysed Smith et al.[51], possessed the standard Dahomey genetic background.

**Experimental design**. Because of the backcrossing scheme used, all experimental females (including *SPR*, *UAS-rpr* and *InsP3GAL* females), and the rival males, were homozygous for *w^1118^*, while focal males possessed the wild-type *white* gene, and were also homozygous for *spa* (Supplementary Figures 1 and 4). Our use of *spa*—which disrupts the eye phenotype for focal males was designed to create a more even competitive field than if we had used fully wild-type focals: white-eyed males have impaired vision and reduced mating performance in daylight (see ref.[52]), but *spa* also likely compromises visual acuity to some extent. Although individual focal (*spa*) males performed, on average, better than the individual rival males in terms of mating success (Supplementary Table 1), collectively the three rivals still gained the clear majority of matings within groups (i.e. the sum of the matings achieved by the three rivals exceeded the single focal male's matings), thus creating an effective competitive field. Crucially the magnitude of the difference between focal and rival male mating success was indistinguishable between experimental and control treatments, and thus the genotypes used did not introduce any systematic bias that would have altered sexual selection across treatments (see Results section). It is impossible to completely exclude the possibility that patterns of sexual selection on

males would have been different if focals had the same mating success as the rivals, although there is no evidence to suggest that any such differences would negate the clear impact of polyandry seen in our study. Our use of the above constructs meant that we were able to determine the paternity of female (but not male) offspring from the groups using eye colour (see Supplementary Figure 1 for details). We controlled larval density to avoid larval crowding effects on adult body size and other sexual phenotypes[75–79]: all experimental flies were raised at a density of ~200 eggs in 75 ml bottles with ~45 ml of standard maize-molasses fly food. Virgin flies were collected on ice anaesthesia within 8 h of eclosion and kept in single-sex vials of 15–20 individuals for 3–4 days with ad libitum yeast prior to the onset of the experiments. Experimental flies were collected randomly from a pool of emergent adults. On the day before the onset of the experiment, while flies were still virgins, we marked each individual with acrylic paint on the thorax, which has been previously shown to have no effect upon fly behaviour (e.g. refs. [77,80,81]). Paint marking ensured that individuals could be identified during behavioural observations. The group assembly of our experimental design was based on Bjork and Pitnick[73] and Morimoto et al.[77]. Replicate vials contained four flies of each sex (i.e. eight flies per vial): three white-eyed (w[1118]) rival males and one red-eyed focal male (which was w+) were housed with either (i) four SPR− females or four white[Dahomey] control females (referred to as 'control[SPR]') for the SPR experiment, or (ii) four mNSC-ablated females or four control females (either UAS-rpr or InsP3GAL) for the ablated female experiment (Supplementary Figure 4). Males and females of both treatments were allowed to interact for 4 h in the mornings (9 a.m.–13 p.m.) followed by 20 h of egg laying (13 p.m.–9 a.m.), over a 4-day period[73,77]. Flies were aspirated without anaesthesia. Egg-laying vials were retained for 15 days allowing ample time for the offspring to develop and emerge for all genotypes. The female offspring were counted and the eye phenotype recorded to determine paternity (Supplementary Figure 1), thus providing measures of reproductive success of the focal males. All offspring were counted as adults 13–15 days after oviposition. There is no evidence to suggest differential offspring sex ratios between control[SPR] and SPR− females (see Supplementary material in ref. [52], sex ratio: $\chi^2$ value = 0.0012, df = 1, $p > 0.971$). Thus, daughter production was an appropriate proxy of reproductive success for both males and females. For the experiment in which we reduced female productivity, we used two sets of controls (henceforth called control[ablated]), which were the daughters of InsP3GAL females and white[Dahomey] males and UAS-rpr were the daughters of white[Dahomey] females and UAS-rpr males. We conducted experiments using the same methods as for the SPR treatments, except replacing SPR− females with UAS-reaper > InsP3-GAL4 females (henceforth mNSC-ablated) females, and using the appropriate controls. The two control genotypes did not differ in the key phenotypes (offspring production, mating frequency; Supplementary Table 7) so were combined into a single control group for the statistical analyses.

**Data analysis.** All analyses were performed in R version 3.2.2[82]. *Male reproduction* —We analysed the response of mating frequencies (i.e. number of matings obtained per individual), number of unique mates (M), repetitive matings (i.e. number of individual matings between a male and the same individual female mate) and offspring (daughter) production (T), while controlling for the effects of experimental replicate, using a generalised liner model (GLM) with a quasipoisson function, to account for the overdispersion of the raw data. We used a GLM with a quasibinomial function to test for the effects of increasing polyandry on the proportion of daughters sired by the focal male (P), again while accounting for overdispersion of the data. The model was fitted using the 'cbind' function with the number of daughters sired by the focal male as first argument, and the number of offspring not sired by the focal male with the female as the second argument of the function (failures). P values were obtained from F statistics. *Opportunity for pre- and post-copulatory sexual selection*—We calculated $I$, $I_S$ and $I_P$, as described in Table 1. To test for differences between treatments for these population-level parameters, we used a bootstrap (package 'boot') to calculate the 95% confidence interval of our estimates. Only when the confidence intervals did not overlap did we consider the differences statistically significant. *Selection gradients and overall strength of pre- and post- copulatory sexual selection*—We used both mean- and variance-standardised multivariate linear model of focal male T on M, P and N (variance-standardised analyses measure s'max, also known as Jones' indices)[15,42,77,83,84]. To calculate the relative importance of increased mating frequency with the same female for male P, we fitted a general linear model of mean-standardised P on the average number of repetitive matings per mate (see Table 1, Supplementary Tables 2–6). We tested the significance of the factors in the model with a quasibinomial GLM. We squared-root boxcox-transformed male reproductive success in models of selection gradients for tests of significance; we confirmed the fit of the model through inspection of the diagnostic plots. While mating success data were available for focal and rival males, our experimental design only allowed for paternity share data to be collected for focal males. Thus, mating success could be mean- and variance-standardised within groups, whereas paternity share could only be standardised between focal males across groups. To ensure indices were comparable, we standardised all sexual selection indices across groups within treatments[2,28,46]. *SCI and non-random mating*—It is increasingly realised that the pattern of mate sharing in a population rather than simply the average polyandry can also affect the strength of selection in males[30,43–45,85]. We calculated a male's SCI as the harmonic mean of male's partners mating success[2,28,46,85,86]. To assess

the relationship between male reproductive success (T) and SCI, we fitted a GLM for each treatment of T on M and SCI with quasipoisson distribution to account for overdispersion of the data, while controlling for vial productivity and experimental replicate. To assess the role of SCI in post-copulatory sexual selection, we fitted a GLM for each with P on SCI using the 'cbind' function, while controlling for vial fecundity and experimental replicate. To investigate the presence of non-random patterns of mate sharing in our population, we calculated the SCIC for each treatment. This measures the slope of the least-square regression between male mate number (M) and the intensity of sperm competition (SCI). Negative values of SCIC describe a correlation for males with high mating success to mate with females with few mating partners and positive values describe a positive correlation between male mating success and the mating success of their mates. Considering groups with similar levels of polyandry, positive values of SCIC should further reduce the correlation between male mating success and reproductive success (i.e. reduce the male Bateman gradient), whereas a negative SCIC values will steepen the slope of the male Bateman gradient by accentuating the relationship between mating success and male reproductive success. Importantly, non-zero values of SCIC are likely to arise through chance[28]. To test whether patterns of SCIC were more or less than could be expected by chance, we used randomisations of our behavioural mating data to test the significance of the relationship between male SCI and M. Briefly, our randomisations randomly shuffle the identity of copulating pairs of males and females within each experimental vial (i.e. who mates with who) while holding constant both the total number of copulating pairs and the variation in male and female mating success within each vial (i.e. controlling for average polyandry and the variance in male and female mating success; ref. [87]). We generated 1000 randomisations of our behavioural mating data for each treatment. For each randomised data set we calculated SCIC as above. The small number of individuals and large number of mating pairs restricted the number of randomised outcomes possible, and in addition, the identity of copulating pairs in some vials could not be shuffled and could take only one possible SCIC value. Nonetheless, at the treatment level across all vials this process generated a null distribution of SCIC values. We then compared our observed values to the simulated null distribution of values for each treatment, respectively, to assess whether observed values were more extreme than expected by chance[88,89]. Calculations of SCIC used data including all males, both focal and non-focal males. In all models SCI and M were standardised by dividing by their respective means.

**The Bateman gradient analysis and univariate pre- and post-copulatory s'max (Jones' Index).** Bateman gradients ($\beta_M$), as with phenotypic selection gradients more generally, can be measured as the slope of the least-squares regression of reproductive success (T), standardised to a mean of 1 on mating success (M) as:

$$\beta_M = \frac{COV_{TM}}{\sigma_M^2}, \qquad (1)$$

where $\sigma_M^2$ is the variance in the mating success. There are two key approaches to standardise selection gradients to allow comparisons across traits, groups, populations and species: mean- and variance-standardised gradients[86,90]. Bateman gradients ($\beta_M$) are typically measured as mean-standardised univariate Bateman gradients ($\beta_M^\mu$)[42] and can be calculated by multiplying by mean mating success as:

$$\beta_M^\mu = \beta_M \bar{M}, \qquad (2)$$

whereas variance-standardised univariate gradient ($\beta_M^\sigma$) can be calculated as

$$\beta_M^\sigma = \beta_M \sigma_M. \qquad (3)$$

The slope of a mean-standardised gradient can be interpreted as the expected change in relative reproductive success with a doubling in mating success, that is, a mean-standardised univariate Bateman gradient of 1, tells us that a 1% change in mating success results in a 1% change in relative reproductive success[86]. Variance-standardised univariate Bateman gradients can be interpreted as the change in relative reproductive success for a change in 1 SD in the mean mating success. These two approaches are related such that the variance-standardised slope can be calculated by multiplying the coefficient variation in mating success ($CV_M = \sigma_M/\bar{M}$) by the mean-standardised univariate Bateman gradient as:

$$\beta_M^\sigma = \beta_M^\mu CV_M. \qquad (4)$$

Importantly, since the $CV_M$ is equal to the square root of the opportunity for sexual selection ($I_S$), that is, $\sqrt{I_S} = \sqrt{\sigma_M^2/\bar{M}^2}$, the variance-standardised univariate selection gradient is equal to the univariate s'max $= \beta_M^\mu \sqrt{I_S}$, that is, the pre-copulatory Jones' index[42]. We used this approach to calculate the univariate pre-copulatory Jones' index (s'max), and replaced relative mating success (M) with relative paternity share (P) to calculate the univariate post-copulatory Jones' index. Multivariate pre- and post-copulatory s'max were calculated as the variance-standardised slope of T on M or P from the multiple regression of T on M, N and P. For detailed discussion of univariate Bateman gradients, Jones' index and their relationship to multivariate Bateman gradients that estimate $\beta_{SS}^{Multi}$ while controlling for the effects of phenotypic traits see Henshaw et al.[39].

**The gradient of adjusted paternity share on repetitive matings**. We used the approach of Devigili et al.[47] to examine the effect of adjusting the paternity share measure based on the number of realised sperm competitors. To do so, we calculated the adjusted paternity share of the focal males with each of the females they mated with as:

$$\text{Adjusted PCS} = \text{PCS}_{\text{obs}}(n-1)/(\text{PCS}_{\text{obs}}(n-2)+1), \qquad (5)$$

where $\text{PCS}_{\text{obs}}$ is the paternity share of the focal male across the females with whom he mated, and $n$ is the number of rival males that mated with those same females. Females that were strictly monogamous (only mated with one rival or focal male) or failed to mate with the focal male were not included in the analysis as in Devigili et al.[47]. We then used the average adjusted paternity share (Adjusted PCS) for each male as the response variable in the model and the average rate of repetitive mating for each male as the explanatory variable. This approach thus asks whether repetitive matings with the same female results on average in higher (or lower) paternity share for males over and above that expected given the number of sperm competitors they face.

**Re-analysis of published data**. We analysed data from Smith et al.[51] to test whether any observed differences between $SPR-$ and control$_{SPR}$ treatments could have been driven by a shift in patterns of sperm storage (Supplementary Figure 9). In their study, Smith et al.[51] set out to investigate how changes in the strength of sexual conflict influenced the reproductive strategies of females as well as her previous and current mates. As in our study, Smith et al.[51] performed experiments in control temperature rooms on a 12:12 light–dark cycle at 25 °C. Their study used the same $SPR-$ and control$_{SPR}$ lines, in the same genetic background, as our study. Flies were collected within 6 h post emergence to guarantee virginity, and were housed in single-sex vials for 3–5 days before the experiments. The authors mated control$_{SPR}$ and $SPR-$ females, and remated these females at different intervals (i.e. 3, 5, 24, and 48 h after mating), after which the paternity share of first and second males was assessed. Females that failed to remate within 2 h from the start of the remating trial were discarded. In our re-analysis here, we randomly selected one male for each mating trio and labelled him as the 'focal male' (with no replacement). We did this for all trios, and across all remating intervals, to create simulated populations of focal males that mated with control$_{SPR}$ and $SPR$-lacking females. We then calculated the standardised variance in male paternity share $I_P$ for each population as done in our study (see Table 1). We repeated this procedure 1000 times in order to estimate whether $SPR$ deletion and changes in sperm storage altered the variance in male paternity share. Differences between mating frequency between focal and rival males were assessed with a GLM quasipoisson model with treatment, male status (i.e. Focal vs. Rival) and their interaction.

**Reporting summary**. Further information on experimental design is available in the Nature Research Reporting Summary linked to this article.

## Data availability

Data are available in the Oxford Research Archive (ORA) [https://doi.org/10.5287/bodleian:J5kpxjJB0]. Data for Smith et al.[51] is available in Dryad [https://doi.org/10.5061/dryad.h5346][91]. A reporting summary for this Article is available as a Supplementary Information file.

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

## Acknowledgements

J.M. was supported by a DPhil scholarship from the Brazilian National Council for Scientific and Technological Development (CNPq 211668-2013/3), T.P. and G.C.M. by an industrial LINK award from the BBSRC and Aviagen® (BB/L009587/1). S.W. was supported by a BBSRC fellowship (BB/K014544/1), and J.C.P. by a fellowship from Jesus College (Oxford) and an NERC Fellowship (NE/P017193/1). T.C. is supported by NERC (NE/J024244/1 and NE/K004697/1).

## Author contributions

J.M., D.T.S., J.C.P, T.C., T.P. and S.W. designed the experiments. J.M., G.C.M., E.S. and D.T.S. conducted the experiments, and collected the data. J.M., G.C.M. and E.S. analysed the data. All authors contributed to the writing of the paper.

## Additional information

**Competing interests:** The authors declare no competing interests.

