## [Peer Review File · Nature Communications]

Reviewers' comments:

Reviewer #1 (Remarks to the Author):

In my estimation, Pizzari, McDonald and colleagues are the research group most responsible for moving the leading edge of sexual selection theory forward these days. In particular, their application of network theory combined with clever empirical approaches to explore and test relationships between episodes of selection in a highly quantitative framework while simultaneously addressing underlying mechanisms (and hence targets of selection) is bringing much-needed momentum to the field. The majority of studies addressing relationships between pre- and post-copulatory sexual selection have merely addressed covariance in male performance between episodes or examined whether traits respectively functioning exclusively in one episode or the other trade-off with one another. Albeit interesting and relevant, these contributions have ignored the forest for the trees. With this manuscript, the forest is experimentally tackled in a model system by a research team with an eye for nuance. The clever experimental approach reported in this manuscript - genetically manipulating the female receptor for the seminal component sex peptide in order to manipulate levels of polyandry, followed by a rigorous quantitative assessment of sexual selection intensity/opportunity/gradient - represents another uniquely valuable contribution to the field. The value comes not only from the use of an experimental approach to test explicit hypotheses about the relationship between polyandry and the relative contributions of pre-SS and post-SS to variance in male reproductive success, but from tracking the behavior (and paternity consequences) of focal individuals (rather than just looking for pattern differences between treatments) so that assortative mating patterns that could influence covariance structure can be discerned.

The manuscript is very clearly written, rather brilliantly so given the complexity of the analyses. For example, section iii of the Results/Discussion - I had to read this three times to parse it all out. I really like the multivariate approach, the really thoughtful way you progressed through the analyses (e.g., transitioning to application of Jones's index), and the clarity of the reporting. I think this clear presentation of operationally applying each of these different SS indexes to your data, along with the way you role out the justification for each and how the results relate to one another, will do more for clarifying the relationships among the indexes for the community than has most of the theoretical papers published to date.

** This manuscript may be as close as I've come in my career to a recommendation of "accept without revision." I have only one concern with the manuscript that gets highlighted in section (v) of the R/D but permeates the study; it's a "saturation of the matrix" concern. The authors discuss this concern, and put something of a silver lining on it by discussing the adaptive value of repeated matings with the same female. But I contend the current discussion does not address this important concern with sufficient veracity, and I hate to see this extraordinary research program tarnished. As it stands, I just cannot shake the niggling feeling that the most important conclusions (reduced intensity of precop SS and enhanced contribution of paternity share to variation in RS in the SPR- treatment) might be, at least in part, an artifact of mate saturation due to the 4 males/4 females design. In nature, it is hard to envision even the opportunity of repeated mating by the same male and female presenting itself to any meaningful degree. As such, the strategy of doing so would unlikely be under selection. I suspect the natural (i.e., controlSPR) remating frequency of melano is such that the matrix will be relatively unsaturated given the 4d duration of these experiments. Even with no mate recognition/random mating, there should be a relatively low frequency of mating with the same female. However, the increased level of polyandry in the SPR- treatment may cross a flexion point of sorts. It seems intuitive that each mating by a male with a previous mate will devalue that mating's contribution to reproductive success (relative to a mating with a novel female); at the same time, each mating by a male to a previous mate will increase that male's paternity share with that female (in contrast to a mating with a novel female). A case needs to be made that decreased importance of precop SS and an increased importance of postcop SS with elevated polyandry is not simply a consequence of this

saturation. Could a simulation be run, maintaining identified levels of assortative pairing etc, that would compare the empirical results with those when no repeated pairings occur?

Other than that, all I can help out with are a few typos:

1. 147: missing word - "genetic markers we used allowed us to ? the paternity of"
2. 353: revise to state "In Smith et al.47, females..."
3. 393: change "the" to "their"

Great study. I'm deeply impressed.

Reviewer #2 (Remarks to the Author):

I very much enjoyed reading this well-written paper of a thoroughly and carefully executed study on the impact of polyandry on the opportunity and sexual selection in males in *Drosophila* fruit flies. The authors exploited the heightened remating frequency of females lacking the sex peptide receptor to experimentally manipulate the relative importance of pre- and postmating sexual selection. Using this approach, the authors provide strong and clear experimental evidence that elevated polyandry can weaken the overall opportunity of sexual selection and shift the relative importance of pre- and postmating sexual selection toward the latter. This is common assumption in the field, but clear data in support have so far been lacking. Thus, I believe this paper will be of great interest to the field.

I don't have any major criticisms, but numerous (mostly minor) suggestions for improving the paper, which I list below by line.

My main concern, however, is the treatment of postcopulatory sexual selection. On line 145, the authors claim that they were able to keep track of the reproductive success of individual females, but on line 409 they say that females were kept in group vials between observation periods. Unless females had individual markers expressed by their offspring (for which I find no evidence), I do not know how tracking of individual reproductive success should be possible. This is not even possible for the females mated to the focal male if there was more than one female. At best the authors were able to monitor the number of matings, but not who laid which eggs (or their number). This further makes me wonder how they calculated paternity shares, which I would find more informative (and correct) at the level of females than at the group level, especially if females varied in their contribution to the pool of progeny for each group. Since virgin females tend to be less selective than previously mated females, and given that control males and females only mated about twice over the 4-day period (thus generating a few days' worth of progeny from the first mating), much seems to be coming down to who mated first. Note that these matings obviously result in paternity, but they are not competitive and so do not really reflect postcopulatory success in the sense of sperm competition. The counter would have to start after females have remated.

Further, it is unclear to me how the authors calculated the variance in paternity (or postmating sexual selection) if for each group they could at best compare the focal male against all others, but not their individual relative contributions. Hence, the variance must have been calculated between groups (among the different focal males?) where males were not in competition among one another. By contrast, the variance of premating selection may have been calculated at the level of individual males in competition, as their identity was trackable. To me, these data are not really comparable directly. What happened among males within a vial was non-independent, but different groups were independent of one another. Can you please clarify and justify why what you did was correct?

More specific comments:

54 While it is true that sexual selection arises if one sex is more limited (in numbers or reproductive potential) and the opposite sex thus tends to compete for access, sexual selection itself is not restricted to intrasexual competition but also creates variation in reproductive success through intersexual preferences (e.g., female choice).

65 add a comma at the end of the line

67 odd wording - I guess you mean either "reduce the strength of the correlation" or "weaken the correlation"

75 hyphen should be dash

84 To my knowledge, "assortative mating" typically refers to non-random mating with regard to genetic similarity or specific phenotypic traits (e.g., body size), with individuals that exhibit more similar characteristics mating more frequently than predicted by chance. Considering mating between males with high mating success and low-polyandry female as being assortative thus seems to rather confuse a specific term than add any relevant information. I suggest avoiding this term throughout the current paper (it's not necessary anyway), or just say "non-random mating," which is more neutral.

92 correct to "takes into account"

103 first "female" needed? I assume that if the SPR is located in the female tract it's a female receptor by definition...

108 Are you sure you're not referring to the premating opportunity of sexual selection (i.e., I_s)? I'm not convinced that elevated polyandry necessarily reduces the total opportunity of sexual selection (i.e., I) – it depends on the relative importance of pre- and postcopulatory sexual selection. Under the assumption that I_s always explains a greater proportion of I , I can see that any increase in I_p may not be able to compensate for a decline in I_s , but if I_p is large enough, a shift toward postcopulatory sexual selection would not necessarily decrease I . Please clarify and support with clear theoretical predictions or state more clearly the conditions under which your prediction holds.

143 it would help to state here how long flies were observed (it's tucked away in the methods and supplementary material but is relevant for context)

145 Again, how did you keep track of the reproductive success of individual females if they were kept in group vials (line 409)?

153 shouldn't this "F-value" be a p-value?

152ff I wouldn't say there was "no difference" between treatments as the values in Table S1 clearly differ quite a bit. I can accept that there was "no statistically significant difference," but it is difficult to judge from that table without information on the error around the means.

156 (and throughout) what do you mean by "Appendix SI"? To me it's either the appendix or the supplementary information. Additionally, why not direct the reader to the corresponding table or figure directly, given the SI is a 31-page document?

176 check parentheses

313 insert "in" after second "increase"

317 insert "are" after "that"

353/423 correct to "Smith et al."

389 female -> females

442 I assume repeated copulations by the same males were counted as separate mating events (in the sense that any mating of a rival acts against the success of the focal male)?

454f I assume the two grants should be combined within the same parentheses?

References:

The references are incredibly chaotic, with many volume and page numbers missing, along with other inconsistencies like abbreviated vs. full journal titles, incorrect journal names, capitalized vs. lower-case article titles – just to name a few examples. Please tidy up and follow the formatting guidelines of the journal.

All results tables throughout paper/suppl. mat:

Please provide sample sizes or degrees of freedom, either in the table directly or at least in the caption.

All bar-plot figures throughout paper/suppl. mat:

It would be far more informative to indicate the actual p-values instead of asterisks, because there's quite a difference between $p=0.01$ and $p=0.049$ (both listed as "**") or between $p=0.051$ and $p=0.99$ (both listed as "n.s.")

Fig. 2C given that progeny were distributed across different mates, it would make more sense to me to say "proportion of mates' daughters"

Fig. 3 note that the asterisks for panels B and C are *** and *, respectively, but the legend defines ** and ***.

Supplementary material:

Fig. S1/S3 please ensure that characters (or male/female symbols?) are not replaced by rectangles and other weird symbols

lines 209/235 italicize "w" and delete one "1" in "w11118"

Table S1 what is the error around these means (e.g. s.d. or s.e.m.)?

Reviewer #3 (Remarks to the Author):

This study manipulates the levels of female multiple mating in *Drosophila melanogaster* taking advantage of genetically manipulated lines in which the female sex peptide receptor, SPR, a main regulator of female remating rates, has been removed, to test whether, as predicted by verbal arguments and hinted by recent experimental findings (e.g., Collet et al 2012), polyandry increases the weight of postcopulatory sexual selection relative to precopulatory sexual selection. The study also investigates whether variation in polyandry levels has the capacity to alter the covariance between precopulatory and postcopulatory components of reproductive success (mating success and paternity share, respectively). The results indicate that polyandry weakens precopulatory sexual selection and intensifies competition among males postcopula, as predicted. Further, the study shows an increase in the frequency of repeated matings by males when the SPR receptor is lacking, which suggests that this behaviour can be an effective response by which

males increase paternity share in the face of strong postcopulatory sexual selection. In addition, the results provide evidence for mating assortment such that males with high mating success also enjoy higher reproductive success due to reduced sperm competition.

In all, the study is well conducted, implements well-controlled assays and experiments that take advantage of useful genetic tools, and provides a comprehensive and convincing test of the role that polyandry plays in shaping sexual selection in each of its two consecutive episodes (pre- and post-mating). The study underscores the importance of focusing on mechanisms based on molecular pathways such as the sex peptide pathway to understand sexual selection, and ultimately informs that regulators of female mating behaviour are key elements underpinning sexual selection. There are, however, some aspects of the study that need attention in my opinion, as explained below.

According to the supplementary material it seems that sperm competition intensity is inferred from observational data (mating success). This is likely to overestimate true SCI (e.g., due to, for instance, sperm depletion in males, failed matings, cryptic female choice, sperm displacement, etc.). On the other hand, SCI inferred from paternity data would probably underestimate SCI to some extent (due to high paternity skew, especially in this system). So, both methods would have pros and cons. It would be good if you briefly indicated if some uncertainty around the estimate of SCI could have affected the results and its interpretations.

Please specify if paternity share was adjusted by the realized number of sperm competitors within each group (e.g., see page 7 of Devigili et al. 2015. Multivariate selection drives concordant patterns of pre- and postcopulatory sexual selection in a livebearing fish. *Nature Communications* 6: 8291). As Devigili et al. indicate, a 50% paternity share for a male competing with just one realized sperm competitor is equivalent to a 25% paternity share for a male competing against three realized sperm competitors. I am aware that establishing the number of realized sperm competitors comes with a lot of uncertainty, for the reasons stated above. For example, in situations of effective sperm competition involving many males only a few could attain paternity due to last male sperm precedence and sperm displacement. But still, it may seem reasonable to control for the realized number of sperm competitors in the absence of information regarding the "missing data". I would perhaps suggest that a set of complementary analysis is run on a random set of sexual selection indexes using adjusted paternity share, and that the results are compared to the current results (if the current results do not use adjustment of paternity share, that is). However, you may have strong opinions on the need for such an analysis, based on, for instance, data in this system (e.g., on the relationship between number of sperm competitors and mechanisms of sperm displacement leading or not leading to strong paternity bias). At least this issue warrants some discussion, perhaps in the Supplementary Material.

The implications of having small group sizes for the results of repeated matings is discussed in the text, but an expansion of the expected effects of variation in group size (and other details of the experimental settings, such as the time the flies are allowed to interact) on other parameters would be welcomed, beyond acknowledging that the effects of increasing polyandry on sexual selection on males would likely change. Of course, a proper assessment of this issue would require additional work and the question is beyond the scope of the present study, but maybe you have some initial expectations and indicating these would be useful for future research.

It is not until the results section that the reader learns that in each group there was a single focal male and, unless I have missed something, it seems that the reader does not know the details concerning the composition of the group at this stage either; this is only evident when going through the results, figures and supplementary material. Yet I think this information is important (see the previous comment). I think the reader would appreciate more explicit information about

the composition of the groups (number of individuals and sex), and also the numbers of groups, early on. Some information on number of offspring (daughters) scored for paternity assignment would be useful too, at least in the supplementary material (I do not exclude the possibility that I missed this information).

Estimating paternity share from assigning paternity through only daughters most likely does not imply any bias. But I think it would be good to reassure the reader that this is the most likely case, based on for instance data on sex ratio in *Drosophila melanogaster*. If there are any data out there ruling out a connection between the SPR receptor and variations in sex ratio that would be useful to be shown.

At some point I was wondering whether the competitive field created to test male reproductive success is expected to produce results that can be extrapolated to natural or different situations (e.g., competition against equals). Then I kept reading and I saw that you actually address this question. You state that the magnitude of the difference between focal and rival male mating success was indistinguishable between experimental and control treatments, and that therefore the genotypes used did not introduce any systematic bias. Further, you indicate that it is impossible to completely exclude the possibility that patterns of sexual selection on males would have been different if focals had the same mating success as the rivals, but that there is no reason to suspect that differences would cast doubt on the results that are presented. Without thinking too much about the problem I generally agree with your view. It occurs to me that the differences would probably relate to the probability of detection of effects, perhaps. In principle, using a background (i.e. rival males) that exhibits consistently low or high mating success to test the mating success of focal males (compared to say, using a normal distribution of competitiveness for the rivals, or rivals with more or less similar mating success to focals) would lead to lower variances in the mating success of the focal males. How that affects the probability of detection of effects could be debatable, and it would actually depend on the distribution of mating success, but in principle I agree this is not a major problem if the magnitude of the difference between focal and rival mating success is similar for both the control and experimental treatments. Nevertheless, on this point I suggest that the description of the differences in mating success between focal and rival males in the supplementary material is clarified. I found confusing to read that spa males perform better but that rivals gain the majority of matings, because to me that would imply that individual rival males are better competitors, on average, than focal males. It was only when I went to Table S1 that I understood what was meant (by the way, stating SEs associated to mean values in Table S1 would be handy).

One of the main potentially serious concerns of the study is that the absence of SPR may be associated to unknown or unwanted effects, in addition to the regulation of female remating, which could affect the interpretation of the results (e.g., the absence of SPR could interfere with paternity bias, for instance). The possibility of correlated effects is more worrying when one considers that the *Df(1)Exel6234* deficiency deletes not only the SPR but also removes several genes with unknown function. However, this concern is addressed in the study. Several potential avenues for alternative explanations are ruled out with persuasive arguments but also with robust data. Collectively, the evidence provided is convincing.

Additional comments:

Please specify whether the binomial models use `cbind`. You talk about modelling proportions, but do not specify whether the models account for sample size from where the proportion is calculated (i.e., using, in the case of P , denominators for the numbers of daughters sired by focal males).

Given that multiple sexual selection indexes are examined and compared, have you considered being more stringent when interpreting differences between treatments (e.g., correction for multiple estimates of p when focusing on p values)? Also, what is exactly "partial overlapping" of 95% CI? How is it measured?

The reference list needs attention. There is a lot of missing information. For instance, there is no information on volume number, page number or paper ID number for references 11, 12, 13, 14, 16, 47, etc. For other references there are page numbers but no volume number information, e.g., reference number 44. Citations also need attention sometimes, for instance, citation of Smith et al in line 423, and in line 353.

L330. Effect instead of affect.

L333. The p in IP is stated in Table 1 and in other sections with capital letter, but not here.

I hope the review is useful.

Paco Garcia-Gonzalez

Morimoto *et al.* "Experimental demonstration that sex peptide receptor-regulated polyandry differentially modulates pre- and post-copulatory sexual selection in *Drosophila*"

We sincerely thank the editor and the three reviewers for their positive, detailed and constructive comments. In addressing these comments, we feel the manuscript has been substantially improved. In summary, to address the reviewers' comments, we:

- Included detailed discussion on the implications of the experimental group set-up and the saturation of the mating matrix to clarify that these factors cannot explain the differences in sexual selection detected between the treatments of our study (raised by Reviewers 1 and 3);
- Clarified passages of the manuscript that were unclear, including sections to explicitly explain how the reproductive success of individual females were assessed and how the calculations of sexual selection metrics between (not within) groups were performed (raised by Reviewer 2);
- Justified the use of first copulations in our estimates of sexual selection metrics, and explain why ignoring the first copulations would introduce biases on estimates of sexual selection (raised by Reviewer 2);
- Explained why calculating *SCIC* from paternity can lead to biases, and therefore justified our approach of calculating *SCIC* from observational data (raised by Reviewer 3);
- Re-analysed published data to show that there is no evidence of differences in offspring sex ratio between the control_{SPR} and SPR- lines used in our study (raised by Reviewer 3);
- Added univariate calculation of the pre- and post-copulatory Jones' index (*s'max*) for completeness, analysis of 'adjusted' paternity share (raised by Reviewer 3), and rectified some inconsistencies in the computation of the dataset. This has not qualitatively changed the key patterns of the study, although it has revealed a slightly stronger effect of our SPR- manipulation on selection on male partner fecundity (*N*) and multivariate pre-copulatory *s'max* than previously reported (graphs highlighted in red in revised draft to facilitate revision).
- Rectified the text and figures in accordance to the corrected dataset, and reviewers' comments regarding typos, grammar, and display of the figures (raised by all Reviewers)

Please find a complete point-by-point response to the reviewers' comments below.

Reviewer #1 (Remarks to the Author):

In my estimation, Pizzari, McDonald and colleagues are the research group most responsible for moving the leading edge of sexual selection theory forward these days. In particular, their application of network theory combined with clever empirical approaches to explore and test relationships between episodes of selection in a highly quantitative framework while simultaneously addressing underlying mechanisms (and hence targets of selection) is bringing much-needed momentum to the field. The majority of studies addressing relationships between pre- and post-copulatory sexual selection have merely addressed covariance in male performance between episodes or examined whether traits respectively functioning exclusively in one episode or the other trade-off with one another. Albeit interesting and relevant, these contributions have ignored the forest for the trees. With this manuscript, the forest is experimentally tackled in a model system by a research team with an eye for nuance. The clever experimental approach reported in this manuscript - genetically manipulating the female receptor for the seminal component sex peptide in order to manipulate levels of polyandry, followed by a rigorous quantitative assessment of sexual selection intensity/opportunity/gradient - represents another uniquely valuable contribution to the field. The value comes not only from the use of an experimental approach to test explicit hypotheses about the relationship between polyandry and the relative contributions of pre-SS and post-SS to variance in

male reproductive success, but from tracking the behavior (and paternity consequences) of focal individuals (rather than just looking for pattern differences between treatments) so that assortative mating patterns that could influence covariance structure can be discerned.

The manuscript is very clearly written, rather brilliantly so given the complexity of the analyses. For example, section iii of the Results/Discussion - I had to read this three times to parse it all out. I really like the multivariate approach, the really thoughtful way you progressed through the analyses (e.g., transitioning to application of Jones's index), and the clarity of the reporting. I think this clear presentation of operationally applying each of these different SS indexes to your data, along with the way you role out the justification for each and how the results relate to one another, will do more for clarifying the relationships among the indexes for the community than has most of the theoretical papers published to date.

** This manuscript may be as close as I've come in my career to a recommendation of "accept without revision."

Response: We are very grateful for these very positive comments.

I have only one concern with the manuscript that gets highlighted in section (v) of the R/D but permeates the study; it's a "saturation of the matrix" concern. The authors discuss this concern, and put something of a silver lining on it by discussing the adaptive value of repeated matings with the same female. But I contend the current discussion does not address this important concern with sufficient veracity, and I hate to see this extraordinary research program tarnished. As it stands, I just cannot shake the niggling feeling that the most important conclusions (reduced intensity of precop SS and enhanced contribution of paternity share to variation in RS in the SPR- treatment) might be, at least in part, an artifact of mate saturation due to the 4 males/4 females design. In nature, it is hard to envision even the opportunity of repeated mating by the same male and female presenting itself to any meaningful degree. As such, the strategy of doing so would unlikely be under selection. I suspect the natural (i.e., controlSPR) remating frequency of melano is such that the matrix will be relatively unsaturated given the 4d duration of these experiments. Even with no mate recognition/random mating, there should be a relatively low frequency of mating with the same female. However, the increased level of polyandry in the SPR- treatment may cross a flexion point of sorts. It seems intuitive that each mating by a male with a previous mate will devalue that mating's contribution to reproductive success (relative to a mating with a novel female); at the same time, each mating by a male to a previous mate will increase that male's paternity share with that female (in contrast to a mating with a novel female). A case needs to be made that decreased importance of precop SS and an increased importance of postcop SS with elevated polyandry is not simply a consequence of this saturation. Could a simulation be run, maintaining identified levels of assortative pairing etc, that would compare the empirical results with those when no repeated pairings occur?

Response: These are important points, which we address in two sections below:

Saturation of the mating matrix: The reviewer suggests a case should be made that the reduced importance of precopulatory sexual selection is not due solely to a saturation of the mating matrix and asks whether simulations could be run without remating that would reveal similar patterns. We can confirm that simulations suggest that results of study are not solely driven by a saturation of the mating matrix as a consequence of the small groups used in this study

In a recent study, McDonald & Pizzari (PNAS, 2018) used simulations to explore the effect of polyandry and non-random mating (i.e. assortative mating; SCIC) on patterns of sexual selection over different population sizes and levels of saturation. These simulations assume no remating between individual males and females and first simulated groups of 10 males and 10 females, and then replicated simulations using much larger groups of 100 males and 100 females (i.e. respectively one and two orders of magnitude larger than the groups used in our current study, 4 males and 4 females). This simulation work explored a range of mating assortments spanning those identified in the present study (i.e. simulated SCIC range = approx. -1.5 to 1.5; range in current study = -0.124 to 0.028) and so is directly relevant to our results.

The simulations showed that while the reduction in the importance of pre-copulatory sexual selection is indeed accelerated in near saturated mating matrices, increases in polyandry emerge as a fundamental driver of these patterns across both low and high levels of matrix saturation. Specifically, Figure 3 A&D of McDonald & Pizzari (2018), which show results for 10 male:10 female group size simulations, demonstrate that increasing polyandry leads to a reduction in the opportunity for selection (I) and precopulatory Jones' Index even when the mating matrix ranges from 20% to 80% saturation. This effect is observed at the SCIC values recorded in the present study. Figures S9 A&D of the same paper, which show 100 male:100 female group size simulations, replicate these

findings where the matrix is as sparse as 2% to 20% saturation, again at similar SCIC values (McDonald & Pizzari, 2018). This shows that reductions in precopulatory sexual selection due to increased polyandry are predicted in both small and large populations, and in both sparse and more saturated mating matrices. Moreover, in our most polyandrous SPR null treatment, the mean female polyandry was <3 male partners (Fig S2A). Thus in a group of 4 males and 4 females, matrices are likely to fall well below the full saturation (i.e. 100% saturation when all 4 females mate with 4 male partners). Thus, while higher levels of saturation of the mating matrix are likely to accelerate the reduction in pre-copulatory sexual selection in our study, the simulations indicate that the effects presented in this study cannot be driven solely by a saturation of the mating matrix, and that polyandry effects on sexual selection are present over a range of group sizes and do not require remating between individual male and female pairs.

We have now addressed this point in detail in the revised manuscript, with explicit reference to the simulation work in McDonald & Pizzari (2018) in the general discussion of the main text (lines 389-401)

Experimental design: It is absolutely true that the fixed 4 males:4 females group design won't represent the full range of natural histories of *Drosophila melanogaster* in the wild. It is also fair to assume that remating with same partner will be more relevant in small or structured populations, and less so in larger, more panmictic populations. However, we note that the probability of remating will be determined by both the opportunity for individual flies to track partners and the relative fitness returns to a male associated with investing in remating with the same female vs in mating with a new female. While our understanding of the mating patterns of natural *D. melanogaster* populations is relatively limited, it is likely that dispersal and mating behaviour will vary considerably. There is, however, evidence that our results may be relevant to at least some *D. melanogaster* populations. First, there is experimental evidence that in lab populations, individuals recognize sexually novel from sexually familiar partners and bias their mating decisions based on these cues (e.g. females can display preferences for previous mates; Tan et al. 2013). Second, males display aggression to defend recently mated females and patches of food from male competitors. This enables males to guard their mates, which results in a higher share of paternity (Baxter et al. 2015). Collectively, these lines of evidence suggested that remating between pairs could play a role in sexual competition in populations of *D. melanogaster*, although we recognise that this area remains ripe for further research.

However, perhaps more importantly, our results have broader relevance, and shed light into the operation of sexual selection in species with high-remating rates and small tight groups. Through the manipulation of the SPR we have experimentally shifted *D. melanogaster* group mating dynamics away from the natural behaviour for that species, which sheds light on general processes that likely occur under elevated promiscuity in other species. Sexual competition in small groups and remating between male and female pairs is common and an important component of sexual competition in many other species in nature. Male remating with the same female has been documented in other insects such as red flour beetles (Lewis 2004), burying beetles (Steiger et al. 2008) and birch catkin bugs (Reinhold et al. 2015). Remating between sexual partners is very common in many birds species (Birkhead & Møller 1992, Collias & Collias, 1996), including species such as the osprey in which a pair can mate up to 97 times within the few days of a single breeding event (Birkhead & Lessells 1988), and is considered a defensive strategy against sperm competition (see Collet et al. 2012 and McDonald et al. 2017 for evidence in red junglefowl). Remating with the same female is frequent in primates (Boinski 1987, Oklander et al. 2014, Wickings et al. 1993), rodents (Dewsbury 1981) and other social mammals (e.g. Cafazzo et al. 2014) and in many other organisms from hermaphroditic snails (Pelissie et al. 2014) to externally fertilising fish (Spence et al. 2013). Even within the genus *Drosophila*, *D. hydei* females remate within an hour and can remate >3 times in a single day (Markow 1996) which presumably creates the potential for same-partner remating, unless dispersal is extremely high. Thus, in addition to shedding light onto the reproductive behaviour of *D. melanogaster* specifically, our results also have potential broad significance. We have added further detail to the discussion of the effects of group size on the operation of sexual selection, as also suggested by reviewer 3, below (lines 402-441).

Other than that, all I can help out with are a few typos:

1. 147: missing word - "genetic markers we used allowed us to ? the paternity of"

Response: We corrected the sentence that now reads: "the genetic markers we used allowed us to assign the paternity of daughters but not sons" line 155-157.

2. 353: revise to state "In Smith et al.47, females..."

Response: We corrected the reference to Smith's data that now reads: "Smith et al. 48". See line 356.

3. 393: change "the" to "their"

Response: Our revised the statement now reads: "Uncovering whether these inter-specific differences are related to the sex peptide pathway, or other genetic pathways that mediate the degree of polyandry, will facilitate our understanding of the mechanisms underlying their evolution via sexual selection." See line 435-438.

Great study. I'm deeply impressed.

Response: We sincerely thank the reviewer for their positive feedback.

Reviewer #2 (Remarks to the Author):

I very much enjoyed reading this well-written paper of a thoroughly and carefully executed study on the impact of polyandry on the opportunity and sexual selection in males in Drosophila fruit flies. The authors exploited the heightened remating frequency of females lacking the sex peptide receptor to experimentally manipulate the relative importance of pre- and postmating sexual selection. Using this approach, the authors provide strong and clear experimental evidence that elevated polyandry can weaken the overall opportunity of sexual selection and shift the relative importance of pre- and postmating sexual selection toward the latter. This is common assumption in the field, but clear data in support have so far been lacking. Thus, I believe this paper will be of great interest to the field.

I don't have any major criticisms, but numerous (mostly minor) suggestions for improving the paper, which I list below by line.

Response: We are very grateful to this reviewer for the positive comments.

My main concern, however, is the treatment of postcopulatory sexual selection. On line 145, the authors claim that they were able to keep track of the reproductive success of individual females, but on line 409 they say that females were kept in group vials between observation periods. Unless females had individual markers expressed by their offspring (for which I find no evidence), I do not know how tracking of individual reproductive success should be possible. This is not even possible for the females mated to the focal male if there was more than one female. At best the authors were able to monitor the number of matings, but not who laid which eggs (or their number). This further makes me wonder how they calculated paternity shares, which I would find more informative (and correct) at the level of females than at the group level, especially if females varied in their contribution to the pool of progeny for each group.

Response: We thank the reviewer for identifying this area of confusion, which was due to a lack of clarity in our manuscript. We can clarify that we did track the fecundity of individual females and are thus able to identify the paternity share of focal males with their individual female partners. We separated females individually for egg-laying, and thus estimated the reproductive success of all females individually, and calculated the paternity share of the focal males with the females they mated. We have now added this detail to the methods of the main text (lines 453-466).

Since virgin females tend to be less selective than previously mated females, and given that control males and females only mated about twice over the 4-day period (thus generating a few days' worth of progeny from the first mating), much seems to be coming down to who mated first. Note that these matings obviously result in paternity, but they are not competitive and so do not really reflect postcopulatory success in the sense of sperm competition. The counter would have to start after females have remated.

Response: We agree with the reviewer that mating order of males is likely to play a key role in shaping the paternity share of males in our experiments. However, while the reviewer suggests that the first copulation that a male has with a virgin female is not competitive in the sense of sperm competition, we interpret this definition as dealing with a more restrictive view of postcopulatory sexual selection, that is concerned specifically with realised sperm competition rather than the risk of sperm competition more broadly. For example, in some cases, a male may be the only male to copulate with a female due to highly effective postcopulatory mate guarding. However, mate guarding is under postcopulatory sexual selection because it responds to the *risk* of future sperm competition, and acts to reduce female polyandry and increase a male's paternity share (see Dougherty et al. 2016) for a recent and detailed discussion).

In *D. melanogaster* males inseminate *sex-peptide* and other molecules, which function as a 'chemical mate guarding' (i.e. to reduce the propensity of females to remate with another male). This competitive mechanism is analogous to other "defensive" sperm competition traits such as behavioural mate guarding or copulation plugs that are also employed by first males to delay or prevent female remating. The adaptive significance of these traits from a postcopulatory sexual selection perspective is to increase the first-male's share of paternity (i.e. the component of reproductive success determined after mating) by reducing the level of sperm competition faced his sperm. Thus, variation in the ability of males to mate with virgin females and to delay female remating, are by definition under postcopulatory sexual selection to the extent to which they affect variance in paternity share. Failing to consider first copulations or neglecting these first copulations in our account of male reproduction would therefore introduce biases against males that are better at acquiring first-mates and at delaying their remating, and exclude this source of variation in postcopulatory competitive success. We therefore argue that including all of the matings in our analysis is the most appropriate approach.

Further, it is unclear to me how the authors calculated the variance in paternity (or postmating sexual selection) if for each group they could at best compare the focal male against all others, but not their individual relative contributions. Hence, the variance must have been calculated between groups (among the different focal males?) where males were not in competition among one another. By contrast, the variance of premating selection may have been calculated at the level of individual males in competition, as their identity was trackable. To me, these data are not really comparable directly. What happened among males within a vial was non-independent, but different groups were independent of one another. Can you please clarify and justify why what you did was correct?

Response: As pointed out by the Reviewer, metrics are not comparable when calculated between and within groups. For this reason, we only calculated the variance in paternity share and mating success of the focal males across groups (not within) so that all the metrics calculated in our study were comparable. We have now clarified this in the main text (line 499-503).

More specific comments:

54 While it is true that sexual selection arises if one sex is more limited (in numbers or reproductive potential) and the opposite sex thus tends to compete for access, sexual selection itself is not restricted to intrasexual competition but also creates variation in reproductive success through intersexual preferences (e.g., female choice).

Response: We have clarified the wording of this sentence, as follows "Sexual selection is a powerful agent of evolutionary change, and arises from individual variation in fitness due to competition between members of one sex for access to fertilisation opportunities with the opposite sex¹⁻³" (lines 53-55).

65 add a comma at the end of the line

Response: We have corrected the text accordingly. See lines 66

67 odd wording - I guess you mean either "reduce the strength of the correlation" or "weaken the correlation"

Response: We have corrected the sentence, which now reads: "This may in turn reduce the strength of the correlation between mating success and the number of offspring sired (i.e., the Bateman gradient)" See lines 68-70

75 hyphen should be dash

Response: Addressed. See line 76

84 To my knowledge, "assortative mating" typically refers to non-random mating with regard to genetic similarity or specific phenotypic traits (e.g., body size), with individuals that exhibit more similar characteristics mating more frequently than predicted by chance. Considering mating between males with high mating success and low-polyandry female as being assortative thus seems to rather confuse a specific term than add any relevant information. I suggest avoiding this term throughout the current paper (it's not necessary anyway), or just say "non-random mating," which is more neutral.

Response: We agree that the term assortative mating can be confusing. We have addressed this by using the recommended term 'non-random' whenever appropriate, and more explicitly referring to the correlation between a male's mating success (polygyny) and the polyandry of his partners whenever

this detail was required, following convention of previous literature (e.g. McDonald & Pizzari 2018 PNAS) (e.g., line 84)

92 correct to “takes into account”

Response: We have corrected the text accordingly (line 94).

103 first “female” needed? I assume that if the SPR is located in the female tract it’s a female receptor by definition...

Response: We have corrected the sentence that now reads: “and the sex-peptide receptor (SPR) in the female’s reproductive tract and nervous system.” See lines 107-108.

108 Are you sure you’re not referring to the premating opportunity of sexual selection (i.e., I_s)? I’m not convinced that elevated polyandry necessarily reduces the total opportunity of sexual selection (i.e., I) – it depends on the relative importance of pre- and postcopulatory sexual selection. Under the assumption that I_s always explains a greater proportion of I , I can see that any increase in I_p may not be able to compensate for a decline in I_s , but if I_p is large enough, a shift toward postcopulatory sexual selection would not necessarily decrease I . Please clarify and support with clear theoretical predictions or state more clearly the conditions under which your prediction holds.

Response: As the reviewer suggests the prediction that the total opportunity of sexual selection (I) will diminish with increased polyandry is built on the expectation that variance in precopulatory mating success (I_s) should decrease and that variance in paternity share (I_p), while increasing in relative importance, is unlikely to compensate for the reduction in I_s as males are forced to share some paternity across their mating partners. This is in part based on results of previous correlational approaches (Collet et al. 2012). Collet et al. (2012) showed that while the overall relative importance of I_p increases the total opportunity for sexual selection (I) decreases, due to a reduction in I_s . However, we agree that this need not be the case. If the variance in postcopulatory competitive ability was very large, it may be possible for a small number of males to dominate reproductive success through very high paternity share (e.g. at the extreme; no variation in mating success but one focal male outcompetes all others in sperm competition). We have clarified the wording of this section to reflect this pertinent point and to focus on the relative contributions of pre- and post-copulatory sources of variance to I (lines 113-118).

143 it would help to state here how long flies were observed (it’s tucked away in the methods and supplementary material but is relevant for context)

Response: We have now included this information in the methods of the main text. The sentence now reads: “Flies were individually paint-marked, and allowed discrete, fully observed, interaction periods of 4 hours over each of 4 successive days followed by egg laying periods for individual females.” (lines 149-152).

145 Again, how did you keep track of the reproductive success of individual females if they were kept in group vials (line 409)?

Response: We have now clarified that females were held individually for egg laying periods (lines 149-152 and 453-458) See also, our edits in response to comment above.

153 shouldn’t this “F-value” be a p-value?

Response: We have corrected the text accordingly (lines 162-163)

152ff I wouldn’t say there was “no difference” between treatments as the values in Table S1 clearly differ quite a bit. I can accept that there was “no statistically significant difference,” but it is difficult to judge from that table without information on the error around the means.

Response: We have corrected the text which now reads: “with no statistically significant differences between the patterns observed in SPR- and control_{SPR} treatments” (lines 161-162). We also included estimates or errors in Table S1 (see responses below and responses to Reviewer 3 in a similar topic).

156 (and throughout) what do you mean by “Appendix S1”? To me it’s either the appendix or the supplementary information. Additionally, why not direct the reader to the corresponding table or figure directly, given the S1 is a 31-page document?

Response: Apologies if this was unclear, we now refer to specific tables, figure or section that the reader should refer to in the Supplementary Information throughout the main text.

176 check parentheses

Response: We have corrected the parentheses accordingly (line 185)

313 insert "in" after second "increase"

Response: We have now removed this sentence to keep the manuscript streamlined.

317 insert "are" after "that"

Response: We have now removed this sentence to keep the manuscript streamlined.

353/423 correct to "Smith et al."

Response: We have corrected the text accordingly throughout.

389 female -> females

Response: We have corrected the sentence, which now reads: "In some insect species, females are monandrous and thus post-copulatory sexual selection on males is absent, while in others females can mate several times a day⁵⁴, which promotes intense post-copulatory sexual selection and weak pre-copulatory selection⁵⁵". See lines 432-435.

442 I assume repeated copulations by the same males were counted as separate mating events (in the sense that any mating of a rival acts against the success of the focal male)?

Response: The first mating between a unique male and a unique female contributes to a male's mating success (i.e. his number of mating partners, M). Further repetitive matings with the same female were recorded and do not contribute to a male's M but do contribute to his average remating rate with his partners. We have clarified this in the Methods section of the main text (e.g. lines 469-471).

454f I assume the two grants should be combined within the same parentheses?

Response: We have corrected the Acknowledgments section so that the two grants were within the same parentheses.

References:

The references are incredibly chaotic, with many volume and page numbers missing, along with other inconsistencies like abbreviated vs. full journal titles, incorrect journal names, capitalized vs. lower-case article titles – just to name a few examples. Please tidy up and follow the formatting guidelines of the journal.

Response: We apologise for this. We have thoroughly revised the reference section and formatted according to the journal guidelines.

All results tables throughout paper/suppl. mat:

Please provide sample sizes or degrees of freedom, either in the table directly or at least in the caption.

Response: We have now provided sample sizes accordingly (e.g. in legend of figure 2 and 3 main text, lines 836, 850, respectively, and Supplemental Tables S1, and S3-S6).

All bar-plot figures throughout paper/suppl. mat:

*It would be far more informative to indicate the actual p-values instead of asterisks, because there's quite a difference between $p=0.01$ and $p=0.049$ (both listed as "**") or between $p=0.051$ and $p=0.99$ (both listed as "n.s.")*

Response: We have edited the figures and have listed the p-values instead of asterisks where possible. The only exceptions are for comparisons of Opportunity of selection indices. As these are "population-level" parameters that need to be calculated using all males, we used Bootstrapped Confidence Interval, and determined significance based on CI overlap. For these exceptions, we have retained the asterisks and n.s. system (e.g., Figs 2E-G).

Fig. 2C given that progeny were distributed across different mates, it would make more sense to me to say "proportion of mates' daughters"

Response: We have adjusted the y-axis of Figure 2C to read "proportion of mate's daughters.

Fig. 3 note that the asterisks for panels B and C are *** and *, respectively, but the legend defines ** and ***.

Response: We now use of p-values in the figures as addressed in the previous comment, we deleted the asterisks in the legend of Figures 2B and 2C, as well as in all other figures where asterisks are no longer necessary.

Supplementary material:

Fig. S1/S3 please ensure that characters (or male/female symbols?) are not replaced by rectangles and other weird symbols

Response: We have re-formatted the figures and replaced symbols by text.

lines 209/235 italicize "w" and delete one "1" in "w11118"

Response: We have corrected the text accordingly. See lines 229 and 259 of the Supplemental material

Table S1 what is the error around these means (e.g. s.d. or s.e.m.)?

Response: We have now edited Table S1 and state that the errors are standard deviations.

Reviewer #3 (Remarks to the Author):

*This study manipulates the levels of female multiple mating in *Drosophila melanogaster* taking advantage of genetically manipulated lines in which the female sex peptide receptor, SPR, a main regulator of female remating rates, has been removed, to test whether, as predicted by verbal arguments and hinted by recent experimental findings (e.g., Collet et al 2012), polyandry increases the weight of postcopulatory sexual selection relative to precopulatory sexual selection. The study also investigates whether variation in polyandry levels has the capacity to alter the covariance between precopulatory and postcopulatory components of reproductive success (mating success and paternity share, respectively). The results indicate that polyandry weakens precopulatory sexual selection and intensifies competition among males postcopula, as predicted. Further, the study shows an increase in the frequency of repeated matings by males when the SPR receptor is lacking, which suggests that this behaviour can be an effective response by which males increase paternity share in the face of strong postcopulatory sexual selection. In addition, the results provide evidence for mating assortment such that males with high mating success also enjoy higher reproductive success due to reduced sperm competition. In all, the study is well conducted, implements well-controlled assays and experiments that take advantage of useful genetic tools, and provides a comprehensive and convincing test of the role that polyandry plays in shaping sexual selection in each of its two consecutive episodes (pre- and post-mating). The study underscores the importance of focusing on mechanisms based on molecular pathways such as the sex peptide pathway to understand sexual selection, and ultimately informs that regulators of female mating behaviour are key elements underpinning sexual selection. There are, however, some aspects of the study that need attention in my opinion, as explained below.*

Response: We thank this reviewer for the positive appraisal of our study.

According to the supplementary material it seems that sperm competition intensity is inferred from observational data (mating success). This is likely to overestimate true SCI (e.g., due to, for instance, sperm depletion in males, failed matings, cryptic female choice, sperm displacement, etc.). On the other hand, SCI inferred from paternity data would probably underestimate SCI to some extent (due to high paternity skew, especially in this system). So, both methods would have pros and cons. It would be good if you briefly indicated if some uncertainty around the estimate of SCI could have affected the results and its interpretations.

Response: This is an interesting point. It is perhaps worth clarifying the rationale of SCI first. SCI measures the number of other males mating with the same female as a key variable emerging from Parkerian theory, i.e. it represents a first base expectation of the maximum number of sires that can share the paternity of a given set of ova all else being equal. Deviations from this first base expectation are interesting because they identify male- or female-driven mechanisms (e.g. differential sperm depletion, sperm fertilising efficiency, semen transfer, sperm storage, as the reviewer rightly points out), which can be potential targets of post-copulatory sexual selection. As such, SCI is not a direct measure of how tough it is for the sperm of a focal male to fertilise a set of eggs. While SCI is clearly linked to this to some degree, it is also clearly distinct (e.g. competing against one male with very competitive semen may be much tougher than competing with several males each with poor

semen). Therefore, the point of SCI is to estimate the number of males mating with the same female within a unit time.

To generate unbiased estimates, we have opted to calculate SCI based on behavioural mating data. As the reviewer points out, there is a clear expectation that SCI calculated from paternity share alone will introduce bias in measurements of SCI. Previous studies have demonstrated that sexual selection measures are heavily biased by inferring mating from paternity data (Collet et al. 2014), and we therefore avoided such approach. Our supplementary results section SCI shows that it is significantly negatively correlated with both male reproductive success (T) and male paternity share (P), suggesting that in this case, SCI represents a good indicator of the actual intensity of sperm competition faced by males. As suggested, we have now added consideration of the uncertainty around SCI, in lines 506-516 in the methods of the main text.

Please specify if paternity share was adjusted by the realized number of sperm competitors within each group (e.g., see page 7 of Devigili et al. 2015. Multivariate selection drives concordant patterns of pre- and postcopulatory sexual selection in a livebearing fish. Nature Communications 6: 8291). As Devigili et al. indicate, a 50% paternity share for a male competing with just one realized sperm competitor is equivalent to a 25% paternity share for a male competing against three realized sperm competitors. I am aware that establishing the number of realized sperm competitors comes with a lot of uncertainty, for the reasons stated above. For example, in situations of effective sperm competition involving many males only a few could attain paternity due to last male sperm precedence and sperm displacement. But still, it may seem reasonable to control for the realized number of sperm competitors in the absence of information regarding the “missing data”. I would perhaps suggest that a set of complementary analysis is run on a random set of sexual selection indexes using adjusted paternity share, and that the results are compared to the current results (if the current results do not use adjustment of paternity share, that is). However, you may have strong opinions on the need for such an analysis, based on, for instance, data in this system (e.g., on the relationship between number of sperm competitors and mechanisms of sperm displacement leading or not leading to strong paternity bias). At least this issue warrants some discussion, perhaps in the Supplementary Material.

Response: In the results of our original manuscript we did not adjust paternity share estimates to the realised number of sperm competitors. This was because the variance decomposition of the opportunity for selection (I) used in this study is based on breaking male reproductive success (T) into its constituent components as $T = M \times N \times P$. Where M is the unique number of females that a male mates with, N is the average fecundity of his mating partners and P is the proportion of all the available ova (i.e. $M \times N$) that he fertilises. Using adjusted paternity share would thus incorrectly estimate male T and invalidate this variance decomposition. Moreover, a key component of the variation in male P is driven by variation in the number of sperm competitors, i.e. through “chemical mate guarding” and thus an adjusted P would overlook this key source of variance in male paternity share.

However, we do agree that adjusted paternity share can provide additional insights in our study. In particular, we see that using an adjusted paternity share would be an appropriate and informative addition to our analysis of the role of remating with the same female in shaping male paternity share. Specifically, assessing the relationship between male average remating rate and his adjusted paternity share would allow us to assess the role of remating with the same female in determining male paternity share above and beyond that expected, given his number of realised sperm competitors. We therefore include additional analysis of male remating rate and adjusted paternity share as Devigili et al. (2015) in supplementary information (lines 202-206) and reference these results in lines 322-325 of the main text. The results showed the same patterns as our original analysis, whereby the remating gradient on the adjusted paternity share was positive in the SPR - but not in the control $_{SPR}$, even though this difference did not reach statistical significance.

The implications of having small group sizes for the results of repeated matings is discussed in the text, but an expansion of the expected effects of variation in group size (and other details of the experimental settings, such as the time the flies are allowed to interact) on other parameters would be welcomed, beyond acknowledging that the effects of increasing polyandry on sexual selection on males would likely change. Of course, a proper assessment of this issue would require additional work and the question is beyond the scope of the present study, but maybe you have some initial expectations and indicating these would be useful for future research.

Response: We agree with the reviewer that a full exploration of repeated matings and groups size is

beyond the scope of this study, but – as suggested – we have now included an expanded discussion of the effects of group size in the main text: see also our response to reviewer 1. See lines 389-426.

It is not until the results section that the reader learns that in each group there was a single focal male and, unless I have missed something, it seems that the reader does not know the details concerning the composition of the group at this stage either; this is only evident when going through the results, figures and supplementary material. Yet I think this information is important (see the previous comment). I think the reader would appreciate more explicit information about the composition of the groups (number of individuals and sex), and also the numbers of groups, early on. Some information on number of offspring (daughters) scored for paternity assignment would be useful too, at least in the supplementary material (I do not exclude the possibility that I missed this information).

Response: We now include the number of groups and the number of flies (and sex) within each group early on in the main text (lines 97-98). We also included the number of offspring scored for paternity measures (line 126).

*Estimating paternity share from assigning paternity through only daughters most likely does not imply any bias. But I think it would be good to reassure the reader that this is the most likely case, based on for instance data on sex ratio in *Drosophila melanogaster*. If there are any data out there ruling out a connection between the SPR receptor and variations in sex ratio that would be useful to be shown.*

Response: We thank the reviewer for this suggestion. Dean *et al* (2012) calculated the average number of male and female offspring of control_{SPR} and SPR- female lines (see Table S1 in Dean *et al.*, 2012, reproduced below). We performed a Chi-squared test of independency on these data and as expected, found no evidence of differences in the average sex ratio of the offspring between control_{SPR} and SPR- female lines.

Data from Table S1 in Dean *et al.*, 2012 *PLoS Genetics*

	Female line	
Offspring sex	control _{SPR}	SPR-
Female	28.2	11.2
Male	25.9	10.1

Pearson's Chi-squared test

Chi-squared value = 0.0012, df = 1, p-value = 0.9715

We included Dean *et al.* 2012 reference, and the Chi-square test information in the Methods section (lines 460-466).

At some point I was wondering whether the competitive field created to test male reproductive success is expected to produce results that can be extrapolated to natural or different situations (e.g., competition against equals). Then I kept reading and I saw that you actually address this question. You state that the magnitude of the difference between focal and rival male mating success was indistinguishable between experimental and control treatments, and that therefore the genotypes used did not introduce any systematic bias. Further, you indicate that it is impossible to completely exclude the possibility that patterns of sexual selection on males would have been different if focals had the same mating success as the rivals, but that there is no reason to suspect that differences would cast doubt on the results that are presented. Without thinking too much about the problem I generally agree with your view. It occurs to me that the differences would probably relate to the probability of detection of effects, perhaps. In principle, using a background (i.e. rival males) that exhibits consistently low or high mating success to test the mating success of focal males (compared to say, using a normal distribution of competitiveness for the rivals, or rivals with more or less similar mating success to focals) would lead to lower variances in the mating success of the focal males. How that affects the probability of detection of effects could be debatable, and it would actually depend on the distribution of mating success, but in principle I agree this is not a major problem if the magnitude of the difference between focal and rival mating success is similar for both the control and experimental treatments. Nevertheless, on this point I suggest that the description of the differences in mating success between focal and rival males in the supplementary material is clarified. I found confusing to read that spa males perform better but that rivals gain the majority of matings, because to me that would imply that individual rival males are better competitors,

on average, than focal males. It was only when I went to Table S1 that I understood what was meant (by the way, stating SEs associated to mean values in Table S1 would be handy).

Response: We have adjusted the text in the supplementary information to clarify this point (lines 67-69-71 in the supplementary information) and added SE estimates to Table S1 (see also our response to similar comment by Reviewer 2 regarding estimates of variation in Table S1).

One of the main potentially serious concerns of the study is that the absence of SPR may be associated to unknown or unwanted effects, in addition to the regulation of female remating, which could affect the interpretation of the results (e.g., the absence of SPR could interfere with paternity bias, for instance). The possibility of correlated effects is more worrying when one considers that the Df(1)Exel6234 deficiency deletes not only the SPR but also removes several genes with unknown function. However, this concern is addressed in the study. Several potential avenues for alternative explanations are ruled out with persuasive arguments but also with robust data. Collectively, the evidence provided is convincing.

Response: We thank the reviewer for positive feedback on addressing this issue.

Additional comments:

Please specify whether the binomial models use cbind. You talk about modelling proportions, but do not specify whether the models account for sample size from where the proportion is calculated (i.e., using, in the case of P, denominators for the numbers of daughters sired by focal males).

Response: We did use 'cbind' for modelling proportions, which allowed us to account for the sample size from where the proportions were calculated. We now clarify the use of 'cbind' in the main text: "We used a GLM with a 'quasibinomial' function to test for the effects of increasing polyandry on the proportion of daughters sired by the focal male (P), again while accounting for overdispersion of the data. The model was fitted using the 'cbind' function with the number of daughters sired by the focal male as first argument, and the number of offspring not sired by the focal male with the female as the second argument of the function (failures)." See lines 476-479.

Given that multiple sexual selection indexes are examined and compared, have you considered being more stringent when interpreting differences between treatments (e.g., correction for multiple estimates of p when focusing on p values)? Also, what is exactly "partial overlapping" of 95% CI? How is it measured?

Response: A more stringent p-value could lead to an increase in type II errors (false negatives) and therefore we prefer to maintain our significance level to the standard 0.05 and our focus on the magnitude of effect sizes. It is important to mention that for the comparison between the I , I_s , and I_p , which are population-level parameters, we used 95% Bootstrap Confidence Intervals that are more conservative than the standard significance level of 0.05, thus minimising the chances of type II errors. We apologise for the "partial overlapping" of 95% confidence interval as this was not used for the analyses of that data. We have now deleted this from the figure legend of Figure 2.

The reference list needs attention. There is a lot of missing information. For instance, there is no information on volume number, page number or paper ID number for references 11, 12, 13, 14, 16, 47, etc. For other references there are page numbers but no volume number information, e.g., reference number 44. Citations also need attention sometimes, for instance, citation of Smith et al in line 423, and in line 353.

Response: We apologise for this and have now corrected the references in accordance to the journal guidelines.

L330. Effect instead of affect.

Response: Text now reads: "This is likely due to the fact that SPR- females produce fewer eggs than controls after mating³⁴, although the strength of this effect seems to be much weaker in studies in which female multiple mating is permitted than in single-mating assays" (lines 334-337).

L333. The p in IP is stated in Table 1 and in other sections with capital letter, but not here.

Response: We have corrected the text accordingly. See line 339

I hope the review is useful.

Paco Garcia-Gonzalez

Response: Very helpful, thank you!

Reference list

- Baxter, C. M., Barnett, R., & Dukas, R. 2015. Aggression, mate guarding and fitness in male fruit flies. *Anim. Behav.* 109: 235-241.
- Birkhead, T. R., & Lessells, C. M. 1988. Copulation behaviour of the osprey *Pandion haliaetus*. *Anim. Behav.*, 36: 1672-1682.
- Birkhead, T. R., & Møller, A. P. 1992. *Sperm competition in birds. Evolutionary causes and consequences*. London, Academic Press.
- Boinski, S. 1987. Mating patterns in squirrel monkeys (*Saimiri oerstedii*): Implications for seasonal sexual dimorphism. *Behav. Ecol. Sociobiol.* 21:13–21.
- Cafazzo, S., R. Bonanni, P. Valsecchi, and E. Natoli. 2014. Social variables affecting mate preferences, copulation and reproductive outcome in a pack of free-ranging dogs. *PLoS ONE* 9.
- Collias, N. E., & Collias, E. C. 1996. Social organization of a red junglefowl, *Gallus gallus*, population related to evolution theory. *Anim. Behav.* 51:1337–1354.
- Dean, R., Perry, J. C., Pizzari, T., Mank, J. E., & Wigby, S. 2012. Experimental evolution of a novel sexually antagonistic allele. *PLoS Genetics*, 8: e1002917.
- Devigili, A., J. P. Evans, A. Di Nisio, and A. Pilastro. 2015. Multivariate selection drives concordant patterns of pre- and postcopulatory sexual selection in a livebearing fish. *Nat. Commun.* 6:8291.
- Dewsbury, D. A. 1981. Effects of novelty on copulatory behavior: the Coolidge effect and related phenomena. *Psychol. Bulletin*, 89: 464-482.
- Dougherty, L.R., Simmons, L.W., & Shuker, D.M. 2016. Post-copulatory sexual selection when a female mates once. *Anim. Behav.*, 116: 13-16.
- Lewis, S. M. 2004. Multiple mating and repeated copulations: effects on male reproductive success in red flour beetles. *Anim. Behav.* 67: 799-804
- Markow, T. A. (1996) Evolution of *Drosophila* mating systems. *Evol. Biol.*, 29: 73-106.
- McDonald, G. C., Spurgin, L.G., Fairfield, E.A., Richardson, D.S., & Pizzari, T. 2017. Pre-and postcopulatory sexual selection favor aggressive, young males in polyandrous groups of red junglefowl. *Evolution*, 71: 1653-1669.
- McDonald, G. C., & Pizzari, T. 2018. Structure of sexual networks determine the operation of sexual selection. *PNAS*, 115: E53-E61; DOI:10.1073/pnas.1710450115
- Oklander, L. I., M. Kowalewski, & Corach, D. 2014. Male reproductive strategies in black and gold howler monkeys (*Alouatta caraya*). *Am. J. Primatol.* 76:43–55.
- Reinhold, K., Engqvist, L., Consul, A., & Ramm, S.A. 2015. Male birch catkin bugs vary copula duration to invest more in matings with novel females. *Anim. Behav.*, 109: 161-166.
- Spence R, Reichard M, & Smith C. 2013. Strategic sperm allocation and a Coolidge effect in an externally fertilizing species. *Behav. Ecol.*, 24: 82-88.
- Steiger S, Franz R, Eggert A-K, & Müller JK. 2008. The Coolidge effect, individual recognition and selection for distinctive cuticular signatures in a burying beetle. *Proc. Roy. Soc. Lond. B*, 275: 1831-1838. DOI: 10.1098/rspb.2008.0375.

Tan, C. K., Løvlie, H., Greenway, E., Goodwin, S. F., Pizzari, T., & Wigby, S. (2013) Sex-specific responses to sexual familiarity, and the role of olfaction in *Drosophila*. *Proc. R. Soc. B*, 280(1771), 20131691

Wickings, E. J., T. Bossi, & Dixson, A. F. 1993. Reproductive success in the mandrill, *Mandrillus sphinx*: correlations of male dominance and mating success with paternity, as determined by DNA fingerprinting. *J. Zool.* 231:563–574.

REVIEWERS' COMMENTS:

Reviewer #1 (Remarks to the Author):

I was deeply impressed with the original version of this manuscript, and am even more so with this revision. I am satisfied that the authors have satisfactorily addressed my concerns and those of the other referees. I had only a one substantive concern with the original ms: that the mate saturation was artifactually inflating the contribution of PSS-mediated variation in reproductive success under higher levels of polyandry. I appreciate the manner in which the authors addressed my concern.

The text has been revised to highlight this issue for readers, with integrated discussion of a simulation recently reported in McDonald & Pizzari (PNAS, 2018) directly addressing the relationships of concern. I contend this heightened treatment serves to make this study of broader significance to taxa with diverse breeding systems.

I recommend that the manuscript be published in NC.

Reviewer #3 (Remarks to the Author):

The issue regarding the potential for the results to be explained by saturation of the mating matrix due has been addressed to satisfaction bringing results from simulations in a previous study (McDonald and Pizzari 2018). These previous results inform that it is highly unlikely that the weakening of precopulatory sexual selection is driven to a large degree by a saturation of the matrix. I am also satisfied with the arguments and new evidence provided by the authors regarding other issues that I raised, for example, relative to the sperm competition index or to the assignment of paternity through daughters, and I generally agree with the justifications and explanations provided by the authors, and with the modifications made to the manuscript. In all this is a much-improved version of the manuscript, and I have no further comments or suggestions to the authors.